Corrected: Author correction

# Increased autophagy in EphrinB2-deficient osteocytes is associated with elevated secondary mineralization and brittle bone

Christina Vrahnas [1,2,12], Martha Blank[1,2], Toby A. Dite [2,3,12], Liliana Tatarczuch [4], Niloufar Ansari [1,2], Blessing Crimeen-Irwin[1], Huynh Nguyen[5], Mark R. Forwood [5], Yifang Hu[6], Mika Ikegame [7], Keith R. Bambery [8], Cyril Petibois [9], Eleanor J. Mackie [4], Mark J. Tobin [8], Gordon K. Smyth [6,10], Jonathan S. Oakhill [2,3,11], T. John Martin [1,2] & Natalie A. Sims [1,2]

Mineralized bone forms when collagen-containing osteoid accrues mineral crystals. This is initiated rapidly (primary mineralization), and continues slowly (secondary mineralization) until bone is remodeled. The interconnected osteocyte network within the bone matrix differentiates from bone-forming osteoblasts; although osteoblast differentiation requires EphrinB2, osteocytes retain its expression. Here we report brittle bones in mice with osteocyte-targeted EphrinB2 deletion. This is not caused by low bone mass, but by defective bone material. While osteoid mineralization is initiated at normal rate, mineral accrual is accelerated, indicating that EphrinB2 in osteocytes limits mineral accumulation. No known regulators of mineralization are modified in the brittle cortical bone but a cluster of autophagy-associated genes are dysregulated. EphrinB2-deficient osteocytes displayed more autophagosomes in vivo and in vitro, and EphrinB2-Fc treatment suppresses autophagy in a RhoA-ROCK dependent manner. We conclude that secondary mineralization involves EphrinB2-RhoA-limited autophagy in osteocytes, and disruption leads to a bone fragility independent of bone mass.

[1] Bone Biology and Disease Unit, St. Vincent's Institute of Medical Research, 9 Princes Street, Fitzroy, Melbourne, VIC 3065, Australia. [2] Department of Medicine, The University of Melbourne, St. Vincent's Hospital, Melbourne, VIC 3065, Australia. [3] Metabolic Signalling Laboratory, St. Vincent's Institute of Medical Research, 9 Princes Street, Fitzroy, Melbourne, VIC 3065, Australia. [4] Department of Veterinary Biosciences, Melbourne Veterinary School, The University of Melbourne, Parkville, VIC 3010, Australia. [5] School of Medical Science and Menzies Health Institute Queensland, Griffith University, Gold Coast, QLD 4222, Australia. [6] Bioinformatics Division, The Walter and Eliza Hall Institute of Medical Research, Parkville, VIC 3010, Australia. [7] Department of Oral Morphology, Graduate School of Medicine, Dentistry and Pharmaceutical Sciences, Okayama University, Okayama 700-8525, Japan. [8] Infrared Microspectroscopy (IRM) Beamline, ANSTO Australian Synchrotron, Clayton, VIC 3168, Australia. [9] University of Bordeaux, Inserm U1029 LAMC, Allée Geoffroy Saint-Hilaire Bat. B2, 33600 Pessac, France. [10] School of Mathematics and Statistics, The University of Melbourne, Melbourne, VIC 3010, Australia. [11] Mary MacKillop Institute for Health Research, Australian Catholic University, Melbourne, VIC 3065, Australia. [12]Present address: MRC Protein Phosphorylation and Ubiquitylation Unit, James Black Centre, University of Dundee, Dundee DD1 4HN, UK. Correspondence and requests for materials should be addressed to N.A.S. (email: nsims@svi.edu.au)

The skeleton is unique because its organic structure is hardened by integration of mineral, allowing it to provide support for locomotion and protection for internal organs. This mineralized bone forms when collagen-containing osteoid, deposited by osteoblasts, accrues hydroxyapatite crystals. The process has two phases: a rapid initiation (primary mineralization), followed by slower mineral accrual (secondary mineralization) that continues until it reaches a maximal level, or the bone is renewed by remodeling[1]. During bone formation, osteoblasts become incorporated in the non-mineralized osteoid and differentiate into osteocytes to form a highly connected network of specialized cells residing within the mineralized matrix[2]. In response to stimuli such as mechanical load, hormones, and cytokines, osteocytes release proteins that both stimulate[3] and inhibit[4] bone-forming osteoblasts. Both osteoblasts and osteocytes express proteins that initiate osteoid matrix mineralization[5], but it is not known how continuing mineral deposition is controlled.

Two related agents that stimulate bone formation are the agonists of the parathyroid hormone (PTH) receptor (PTH1R): PTH and PTH-related protein (PTHrP). Locally derived PTHrP, from osteoblasts and osteocytes, stimulates bone formation in a paracrine manner[6]. This PTH1R-mediated action is exploited by the pharmacological agents teriparatide (PTH) and abaloparatide (modified N-terminal PTHrP), two agents currently available that increase bone mass in patients with fragility fractures[7,8]. The contact-dependent signaling molecule EphrinB2 is substantially increased in osteoblasts by both PTH and PTHrP treatment in vitro and in vivo[9]. We have previously shown that EphrinB2 (gene name: Efnb2) interacts with its receptor, EphB4, to provide a checkpoint through which osteoblasts must pass to reach full maturity[10,11]. Furthermore, in adult bone, osteoblast-specific EphrinB2 deletion compromised bone strength by impairing osteoblast differentiation and delaying initiation of bone mineralization, ultimately leading to a mild osteomalacia (high osteoid content)[11]. This indicated that osteoid mineralization is initiated by mature osteoblasts beyond the EphrinB2:EphB4 differentiation checkpoint.

We were intrigued by the high level of EphrinB2 expression in fully embedded osteocytes[9], beyond the differentiation checkpoint. Given the extensive connections between osteocytes[2] and the contact-dependent nature of EphrinB2:EphB4 signaling, we hypothesized that EphrinB2 might regulate osteocyte function in the bone matrix. We undertook the present work to determine the requirement for EphrinB2 expression in osteocytes. We found that EphrinB2 in osteocytes does not regulate initiation of bone mineralization, but limits secondary mineral accrual and retains bone matrix flexibility. We also showed that osteocytes lacking EphrinB2 have modified genes associated with autophagy and increased autophagosomes, and that EphrinB2 suppresses autophagy in a RhoA-ROCK-dependent manner; this provides evidence that autophagic processes in osteocytes may directly control mineralization.

## Results

**PTH and endogenous PTHrP increase EphrinB2 in osteocytes.** Upregulation of Efnb2 messenger RNA (mRNA) in osteocytes by PTH and PTHrP stimulation was confirmed in the Ocy454 osteocyte cell line (Fig. 1a); PTH(1–34) and PTHrP(1–141) both significantly increased Efnb2 mRNA levels. Short hairpin RNA (shRNA) knockdown of PTHrP (Pthlh) in Ocy454 cells resulted in significantly lower Efnb2 mRNA levels at all timepoints, compared to vector control (Fig. 1b). This indicates that endogenous PTHrP expression by osteocytes is required for normal Efnb2 expression in osteocytes.

**EphrinB2 deletion in osteocytes causes brittle bones.** To confirm specific Efnb2 deletion, green fluorescent protein (GFP)-tagged osteocytes were fluorescence-activated cell sorting (FACS) purified from Dmp1Cre.DMP1tg-GFP.Efnb2f/f mice. PCR with primers targeted to the Efnb2 exon 1–2 boundary confirmed effective Efnb2 targeting in osteocytes from Dmp1Cre.Efnb2f/f mice, while retaining Efnb2 in non-osteocytic cells (Fig. 1c). Efnb2 was present in GFP-tagged osteocytes from control mice.

Three-point bending tests demonstrated a whole bone strength defect in adult female Dmp1Cre.Efnb2f/f femora compared to Dmp1Cre controls (Fig. 1d–i, Supplementary Table 1), but not in males (Supplementary Table 2). For this reason, all further analysis is focused on the female bones. A significantly higher percentage of bones from female Dmp1Cre.Efnb2f/f mice fractured at lower deformation than controls (Fig. 1e). Average load–deformation curves showed this also (Fig. 1d), and femora from Dmp1Cre.Efnb2f/f mice could withstand less force before yielding (yield force, Supplementary Table 1) and before breaking (ultimate force, Fig. 1f), and deformed less at the maximum force that they could withstand (ultimate deformation, Fig. 1h), compared to controls. The low ultimate deformation was largely explained by a deficit in deformation from the yield point onwards (post-yield deformation, Fig. 1h, Supplementary Table 1). Energy absorbed to failure was also significantly lower in Dmp1Cre.Efnb2f/f femora compared to controls (Fig. 1i). Stiffness (slope of the load–deformation curve before the yield point) was not significantly modified (Supplementary Table 1). This indicates a lower yield point in Dmp1Cre.Efnb2f/f femora, and once they began to yield, they withstood less force and deformed less than controls.

The impaired Dmp1Cre.Efnb2f/f femoral strength was not associated with any significant difference in moment of inertia (Fig. 2a) or cortical bone dimensions (periosteal perimeter and marrow area, Supplementary Table 1). Indeed, when data were corrected for cortical dimensions, impaired strength was still observed. This confirmed that the defect in strength was not due to modified bone shape. Femora from Dmp1Cre.Efnb2f/f mice showed impaired material strength compared to controls (Fig. 2b), including lower yield stress and strain (Supplementary Table 1), lower ultimate stress (Fig. 2c), lower ultimate strain (Fig. 2d), and lower toughness (Fig. 2e). Elastic modulus was not significantly altered (Supplementary Table 1), consistent with the normal stiffness. Reference point indentation (RPI) also indicated a change in material strength independent of geometry, with a significantly greater indentation distance increase (the increase in the test probe's indentation distance in the last cycle relative to the first cycle) in Dmp1Cre.Efnb2f/f bones compared to controls (Fig. 2f). No other parameters measured by RPI were altered (Supplementary Table 3). Dmp1Cre.Efnb2f/f femoral fragility is therefore due to a change in the bone material itself, likely a defect in bone material composition.

**Normal bone formation rate in Dmp1Cre.Efnb2f/f bones.** In contrast to OsxCre.Efnb2f/f mice (with EphrinB2 deleted in osteoblasts), which exhibited reduced mineral appositional rate at the periosteum[11], Dmp1Cre.Efnb2f/f mice exhibited no significant change in periosteal mineral appositional rate (Supplementary Table 1) or periosteal mineralizing surface (Supplementary Table 1). We also assessed trabecular bone structure, which was normal (Supplementary Table 4), and detected no significant difference in trabecular bone formation rate, mineral appositional rate, or mineralizing surface (Supplementary Table 5). There were also no differences detected in osteoid thickness, osteoid surface, osteoblast surface, or osteoclast surface in Dmp1Cre.Efnb2f/f bones compared to controls (Supplementary Table 5). Osteoblast

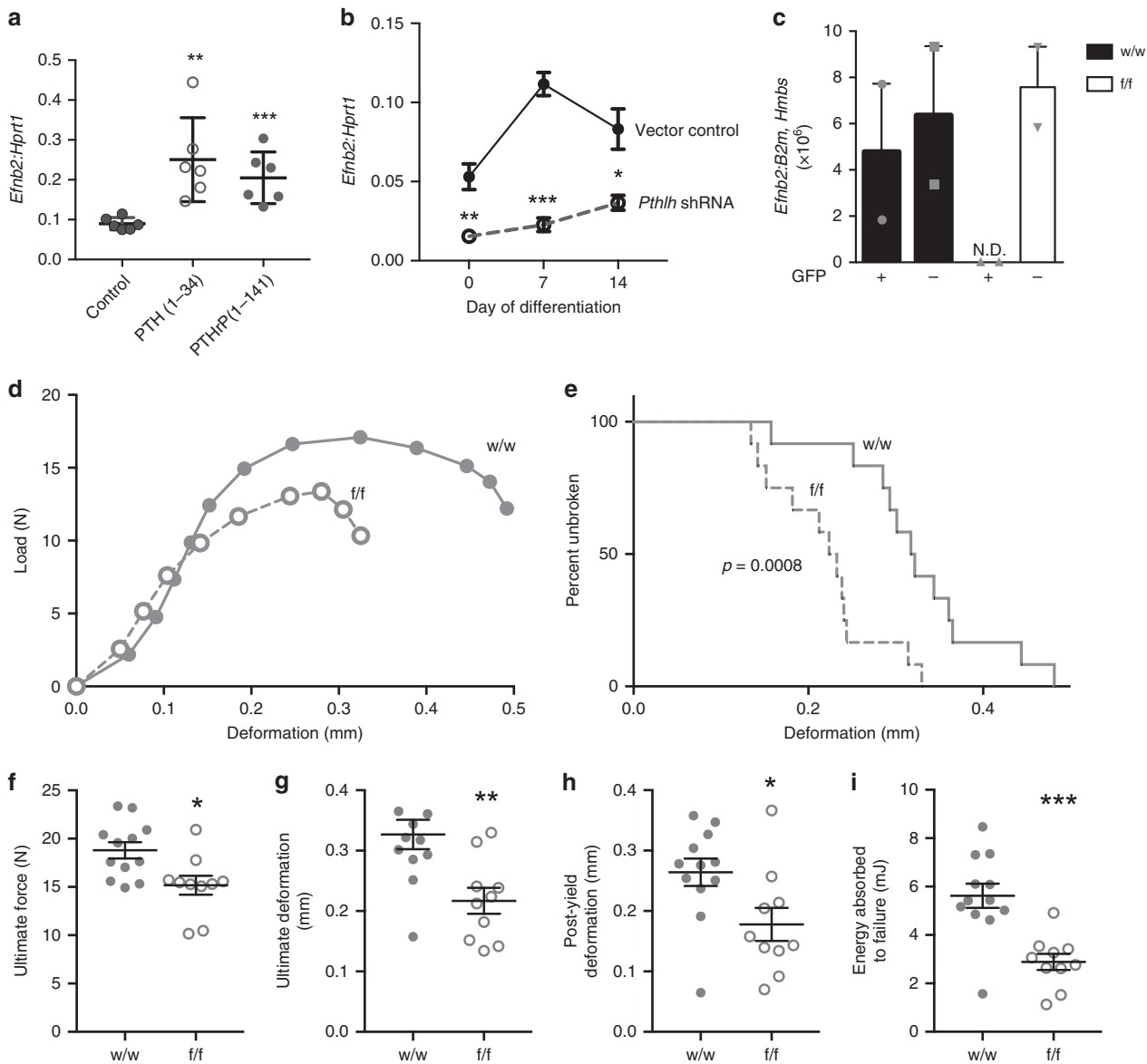

**Fig. 1** Parathyroid hormone (PTH) and PTH-related protein (PTHrP) stimulate *Efnb2* in osteocytes and bone strength is impaired in mice lacking EphrinB2 in osteocytes. **a, b** *Efnb2* messenger RNA (mRNA) levels, relative to *Hprt1*, in Ocy454 cells differentiated for 14 days and treated with 10 nM hPTH(1–34) or hPTHrP(1–141) for 6 h (**a**) and Ocy454 cells with stable short hairpin RNA (shRNA) knockdown of PTHrP (*Pthlh*) at 0, 7, and 14 days of differentiation (**b**). Data are mean ± SEM, $n = 6$ replicates; representative of three independent experiments; *$p < 0.05$, **$p < 0.01$, ***$p < 0.001$ compared to control by Student's $t$ test. **c** *Efnb2* mRNA levels in green fluorescent protein-positive (GFP+) osteocytes isolated from *Dmp1Cre.DMP1-GFP-Tg.Efnb2^{w/w}* (w/w) and *Dmp1Cre.DMP1-GFP-Tg.Efnb2^{f/f}* (f/f) mice compared to GFP− cells. ND = not detected. Data mean ± SEM of two experiments, $n = 6$ mice/group; pooled. **d** Average load–deformation curve from three-point bending tests (each dot represents the average for the noted sample group; error bars excluded to highlight the shape of curves) of femora from 12-week-old female *Dmp1Cre.Efnb2^{f/f}* (f/f) mice and *Dmp1Cre* controls (w/w). **e** Kaplan–Meier curve showing percentage of unbroken femora with increasing deformation ($p$ value from Mantel–Cox log-rank test). **f–i** Calculated indices of bone strength: ultimate force (**f**), ultimate deformation (**g**), post-yield deformation (**h**), and energy absorbed to failure (**i**). Data are mean ± SEM, $n = 10–12$/group. *$p < 0.05$, **$p < 0.01$, ***$p < 0.001$ vs. w/w controls by Student's $t$ test

differentiation, osteoblast-mediated osteoid deposition, and the initiation of bone mineralization therefore occur at a normal rate in *Dmp1Cre.Efnb2^{f/f}* bones.

Backscattered electron microscopy detected no change in osteocyte lacunar size in *Dmp1Cre.Efnb2^{f/f}* femoral midshafts; however, osteocyte lacunar density was significantly greater compared to controls (Supplementary Table 6). This suggests that osteocytes are incorporated more rapidly into the bone matrix during osteoid deposition. No defect in the osteocyte lacuno-canalicular network was detected by Ploton silver staining (Supplementary Fig. 1).

**Greater mineral deposition in *Dmp1Cre.Efnb2^{f/f}* bone**. Since material strength was impaired and the (blinded) technician cutting the sections for histomorphometry noted difficulty in cutting many samples, we assessed bone material by multiple methods. We detected no difference in the proportion of woven to lamellar cortical bone by polarized light microscopy (Supplementary Table 6) and no significant alteration in cortical tissue mineral density (Ct.TMD) measured by micro computed tomography (Supplementary Table 1) in *Dmp1Cre.Efnb2^{f/f}* femora compared to controls. We then used synchrotron-based Fourier-transform infrared microspectroscopy (sFTIRM) to measure bone

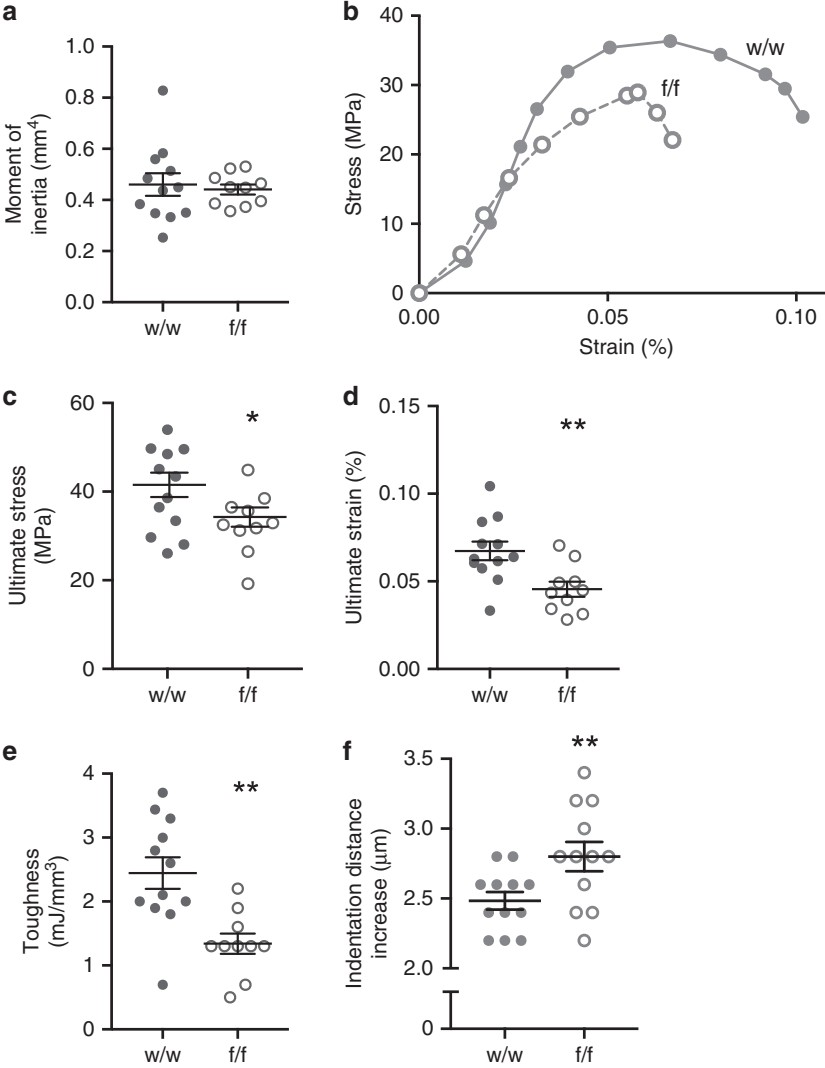

**Fig. 2** Impaired strength in *Dmp1Cre.Efnb2^{f/f}* mice is a material defect and is not caused by altered bone geometry. **a** Femoral moment of inertia. **b–e** Three-point bending test data corrected for bone cross-sectional area, including average stress–strain curve (each dot represents the average load and deformation for the noted sample group; error bars excluded to highlight the shape of curves) (**b**), ultimate stress (**c**), ultimate strain (**d**), and toughness (**e**) of 12-week old female *Dmp1Cre* (w/w) and *Dmp1Cre.Efnb2^{f/f}* (f/f) femora. **f** Indentation distance increase derived from reference point indentation. Data are represented as mean ± SEM, n = 10–12/group. *p < 0.05, **p < 0.01 vs. w/w controls by Student's *t* test

composition at the medial tibial periosteum. Unlike the endo-cortical surface, which is remodelled, the periosteal surface is a site of modelling-based cortical expansion and undergoes continuous bone formation (bone growth)[12]. We have confirmed this in control mice for this study at the site used for sFTIRM analysis at both 6 and 12 weeks of age (Supplementary Fig. 2). Measurements taken at increasing depth thereby allow assessment of bone matrix mineralization rate as it progresses[13,14]. Three regions with increasing matrix maturity were measured (Fig. 3a).

Average spectra for each genotype, taken from the intermediate region, indicated altered spectral geometry in *Dmp1Cre.Efnb2^{f/f}* bone compared to controls (Fig. 3b). *Dmp1Cre.Efnb2^{f/f}* bone had higher phosphate and carbonate peaks, indicating a greater mineralization level. In addition, *Dmp1Cre.Efnb2^{f/f}* bone showed a lower amide I, but higher amide II peak, suggesting greater collagen compaction in *Dmp1Cre.Efnb2^{f/f}* bone.

When quantified, control bones showed the changes associated with bone matrix maturation previously reported on murine periosteum: with increasing depth from the bone edge, mineral:matrix ratio increases, while amide I:II ratio decreases[13]

(Fig. 3c–e). Mineral accrual was accelerated in *Dmp1Cre.Efnb2^{f/f}* bone (Fig. 3c): in the two most immature regions, mineral:matrix ratio was significantly greater in *Dmp1Cre.Efnb2^{f/f}* bone compared to control (Fig. 3c) and amide I:II ratio was significantly lower (Fig. 3e). In addition, although carbonate:mineral ratio did not increase significantly with maturation in the control mice, the carbonate:mineral ratio increased with increasing matrix maturity in the *Dmp1Cre.Efnb2^{f/f}* bone and reached a higher carbonate:mineral ratio than controls in the two more mature regions (Fig. 3d). The greater mineral:matrix ratio was not due to a lower bone collagen content; hydroxyproline levels were significantly higher in whole bone samples from *Dmp1Cre.Efnb2^{f/f}* bone compared to control (Supplementary Table 6). Collagen cross-linking did not change with bone maturity, but was significantly greater in the *Dmp1Cre.Efnb2^{f/f}* intermediate region (Fig. 3f), indicating that crosslinks mature in the same region in which mineral and carbonate accumulate. Male *Dmp1Cre.Efnb2^{f/f}* femora showed no significant modification in mineral:matrix, carbonate:mineral, or amide I:II ratio (Supplementary Fig. 3). The female *Dmp1Cre.Efnb2^{f/f}* brittle bone phenotype is therefore associated

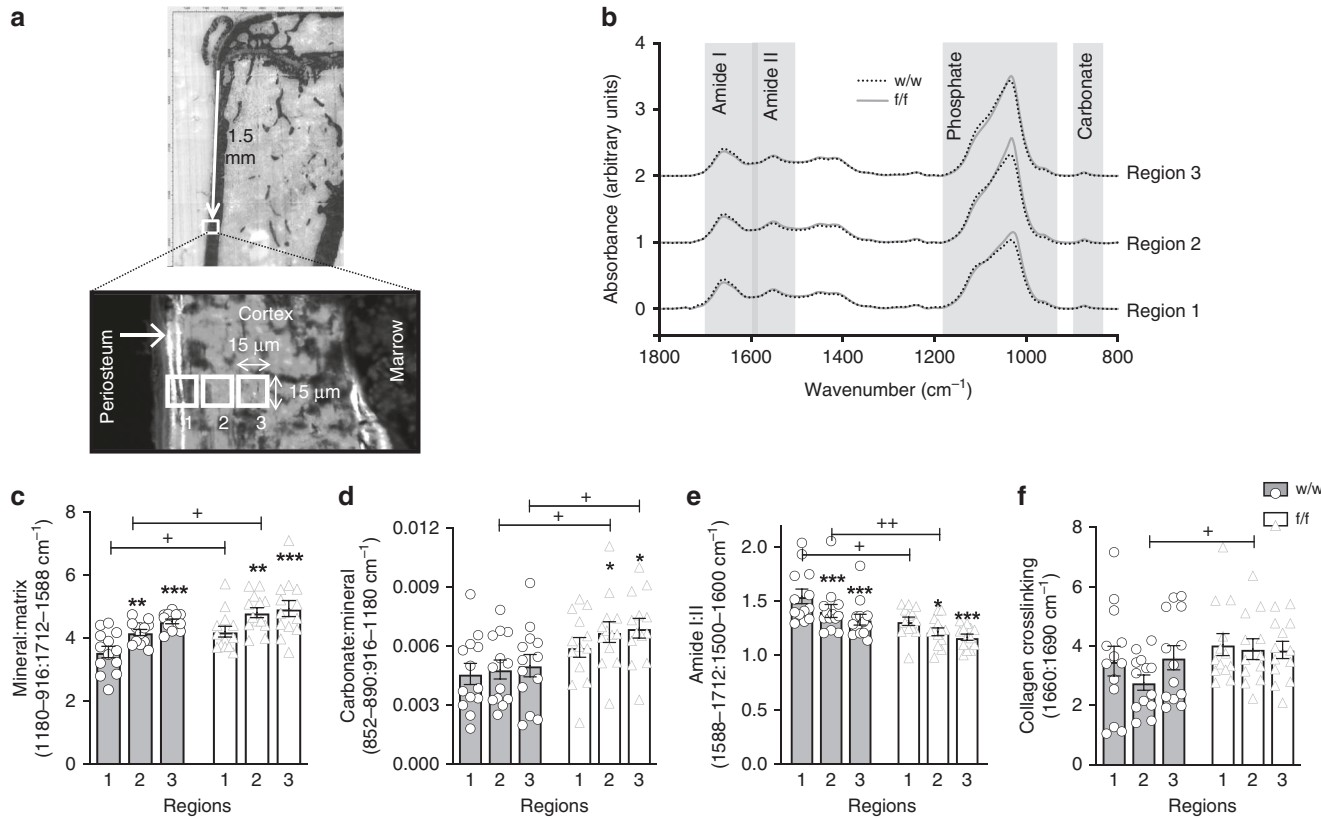

**Fig. 3** Elevated mineral accrual, carbonate deposition, and reduced amide I:II ratio in *Dmp1Cre.Efnb2^{f/f}* cortical bone. **a** Regions of periosteal bone used for synchrotron-based Fourier-transform infrared microspectroscopy (sFTIRM) analysis. Note the double calcein label on the newly mineralized periosteum; regions 1–3 (15 μm² in size) denote bone areas of increasing maturity with increasing distance from the periosteal edge. **b** Averaged sFTIRM spectra from regions 1, 2, and 3 of all 12-week old female *Dmp1Cre* (w/w) and *Dmp1Cre.Efnb2^{f/f}* (f/f) tibiae; grey boxes show approximate regions of the amide I, amide II, phosphate, and carbonate peaks (see Method for details). **c–f** sFTIRM-derived mineral:matrix (**c**), carbonate:mineral (**d**), amide I:II (**e**), and collagen crosslinking (**f**) ratios in regions 1–3 of w/w and f/f tibiae. Data are represented as mean ± SEM, with individual values, $n = 13$/group. $*p < 0.05$, $**p < 0.01$, $***p < 0.001$ vs. region 1 of same genotype (bone maturation effect), $+p < 0.05$, $++p < 0.01$ vs. w/w in the same region (genotype effect) by two-way analysis of variance (ANOVA)

with more rapid bone matrix maturation, including rapid mineral accumulation and carbonate substitution, and more rapid collagen compaction within the matrix.

Since the amide I:II ratio has not yet been validated in the bone as an indicator of collagen compaction, we used polarized Fourier-transform infrared imaging (pFTIRI) in the same region to visualize and quantitate collagen fiber spatial variation (Fig. 4a, b, d, e). We used 0° and 90° polarizing filters to preferentially enhance the signal from molecular bonds oriented parallel and perpendicular, respectively, to the bone tissue section surface[15]. The pFTIRI 0° polarizing filter indicated higher amide I:II ratio in both periosteal and endocortical regions compared to the central cortical bone region (Fig. 4a, c). *Dmp1Cre.Efnb2^{f/f}* bones showed a greater amide I:II ratio in the endocortical region compared to periosteal and central cortical bone (Fig. 4b, c). When quantified, the pFTIRI 0° polarizing filter result confirmed a significantly lower amide I:II ratio in the periosteal region in *Dmp1Cre.Efnb2^{f/f}* bone compared to control, validating our sFTIRM observations (Fig. 4c). The difference in amide I:II ratio between periosteal and central cortical regions was no longer detected due to this lowering in the periosteal region: both showed a significantly lower amide I:II ratio than the endocortical region (Fig. 4b, c). There was no reduction in amide I:II ratio on the endocortical surface; since this region is remodelled, the bone in this region would contain both old and new bone, which may mask any difference. When quantified under 90° polarization, in both control and *Dmp1Cre.Efnb2^{f/f}* mice, amide I:II ratio was lower at

central and endocortical regions due to greater amide II signal (Fig. 4d–f); no difference was observed in fibers aligned under this filter between genotypes (Fig. 4f). This analysis validated the sFTIRM observations, and confirmed a significantly lower amide I:II ratio in the more periosteal *Dmp1Cre.Efnb2^{f/f}* bone than control.

**Dysregulated autophagy genes in *Dmp1Cre.Efnb2^{f/f}* bone**. To identify mechanisms by which EphrinB2 deficiency alters mineral and matrix composition leading to fragile bones, RNA-sequencing was performed on marrow-flushed femora from 12-week-old *Dmp1Cre.Efnb2^{f/f}* and control mice. This revealed 782 up-regulated genes and 1024 down-regulated genes (false discovery rate <0.05). Known osteocyte-specific and mineralization genes were not differentially expressed between the genotypes (e.g., *Dmp1*, *Mepe*, *Sost*, *Phospho1*, *Enpp1*, *Enpp2*, *Fgf23*). Table 1 shows the top 30 differentially expressed genes; none have previously been associated with osteocyte function, bone mineralization, or EphrinB2 function. We noted that a third of the top genes are associated with autophagic processes, specifically with mitophagy and ER-phagy. Six were up-regulated in the EphrinB2-deficient bone: *Fam134b*[16], *Fbxo32*[17], *Lama2*[18], *Bnip3*[19], *Trim63*[20], and *Peg3*[21]. The other four were down-regulated: *Eps8l1*[22], *Klf1*[23], *Tspo2*[24], and *Unc5a*[25]. Genes associated with canonical degradative macroautophagy (*Atg* genes) were not significantly modified.

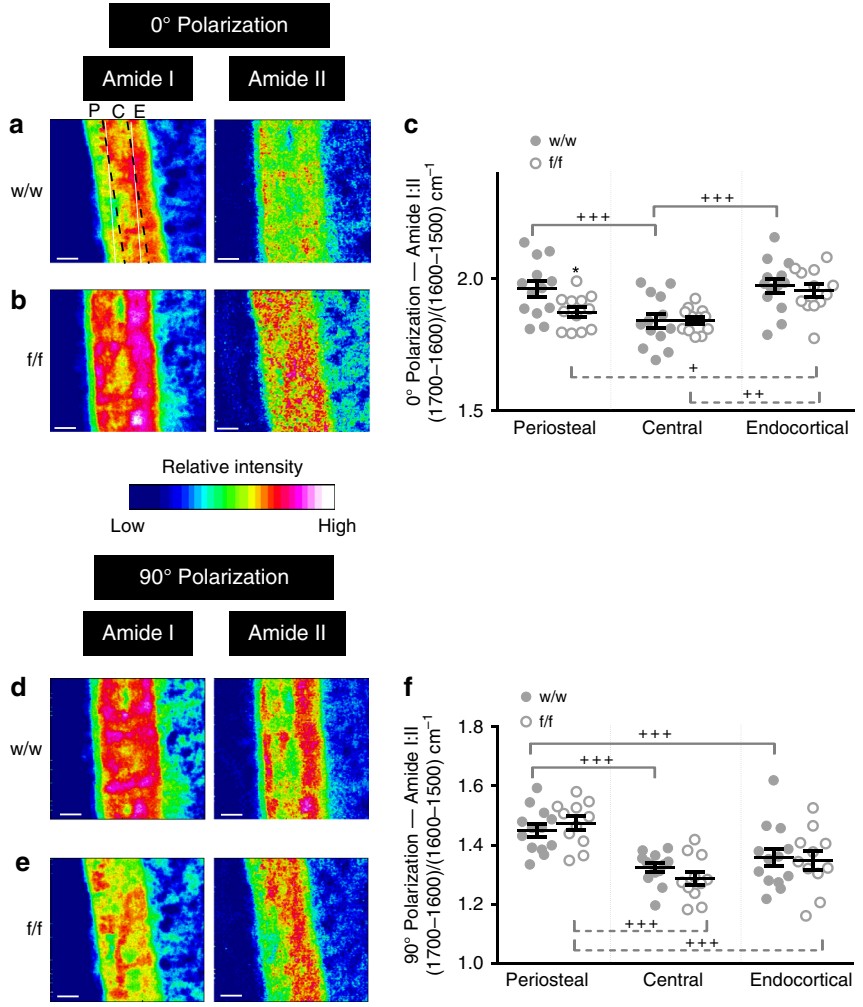

**Fig. 4** Polarized Fourier-transform infrared (FTIR) imaging confirms altered collagen distribution in *Dmp1Cre.Efnb2^f/f* mice. Representative FTIR images showing relative intensity of amide I and amide II peaks in the cortical mid-shaft from 12-week old female *Dmp1Cre* (w/w) and *Dmp1Cre.Efnb2^f/f* (f/f) mice under the (**a**, **b**) 0° and (**d**, **e**) 90° polarizing filters; quantification of the amide I:II ratio under the 0° (**c**) and 90° (**f**) polarizing filters in periosteal (P), central (C) and endocortical (E) regions (shown in the top left of panel a). Data are mean ± SEM, $n = 12$–13/group. *$p < 0.05$ vs. *Dmp1Cre* at the same region (genotype effect), +$p < 0.05$, ++$p < 0.01$, +++$p < 0.001$ vs region indicated by the bracket, within the same genotype (region effect) by two-way analysis of variance (ANOVA). Scale bar = 50 µm

**High autophagy and mineral in *Efnb2*-deficient osteocytes**. To identify whether EphrinB2 deficiency in osteocytes leads to increased autophagy in an independent cell-specific model, we generated *Efnb2*-deficient Ocy454 osteocytes and assessed autophagy levels. Stable *Efnb2* knockdown was confirmed in mature Ocy454 cells at days 11 and 14 (Fig. 5a).

Autophagy was measured in these cells by two methods. First, the autophagosome marker LC3 was detected by immunofluorescence. *Efnb2* knockdown cells (both shRNA 1 and shRNA 2) demonstrated significantly more LC3 punctae than vector control cells (Fig. 5b, c), indicating more autophagosomes in these cells at basal conditions. In control cells, as expected, LC3 punctae were elevated with bafilomycin A1, used as an autophagosome degradation inhibitor to detect autophagosome formation[26] (Fig. 5b, c). In *Efnb2* knockdown cells with bafilomycin A1 treatment, LC3 punctae were not increased further (Fig. 5b, c). This may be because it was not possible to induce a further increase in autophagosome formation; indeed, since *Efnb2* knockdown cells exhibited large accumulated punctae, a further increase would be technically difficult to quantify. Although this method detected more autophagosomes in the basal state in *Efnb2* knockdown cells, it cannot distinguish whether this reflects

more autophagosome formation or less autophagosome degradation.

To assess this specifically, we then assessed LC3-I to -II lipidation by Western blot of cells treated with an alternative lysosomal inhibitor, chloroquine. Chloroquine treatment significantly elevated LC3-II:I ratio in both vector and *Efnb2* knockdown cells (Fig. 5d, Supplementary Fig. 4). This effect was amplified in *Efnb2* knockdown cells. EphrinB2-deficient osteocytes therefore form more autophagosomes than vector control cells (Fig. 5d).

To determine whether the increased autophagy is associated with increased mineralization, *Efnb2* knockdown Ocy454 cells were grown for 2 weeks under mineralizing conditions. Cells stably transfected with shRNA 2 did not survive mineralizing conditions, but the cells expressing shRNA 1, like control cells, remained confluent under mineralizing conditions, and showed greater mineralization than vector control cells (Fig. 5e, f). This provides independent evidence that EphrinB2-deficient osteocytes have both a higher basal autophagy level and greater mineralizing capacity than wild-type osteocytes.

To provide proof-of-principle evidence that increasing autophagy can augment secondary mineralization by osteocytes, we

**Table 1 Upregulation of autophagy-related genes by RNA-sequencing of _Dmp1Cre.Efnb2_<sup>f/f</sup> bone**

| Rank | Gene name | Gene ID | Log 2 FC | Avg Exprn | T | P value | FDR |
|---|---|---|---|---|---|---|---|
| 1 | Ccdc92b | 432582 | −3.679 | 2.509 | −17.73 | 4.23E-08 | 0.00063 |
| 2 | BC049730 | 232972 | −5.568 | 2.260 | −11.20 | 1.92E-06 | 0.00948 |
| 3 | Pira6 | 18729 | 1.788 | 2.245 | 10.52 | 3.19E-06 | 0.00948 |
| 4* | **Fam134b** | 66270 | 1.907 | 6.821 | 10.51 | 3.22E-06 | 0.00948 |
| 5* | **Fbxo32** | 67731 | 2.443 | 7.223 | 10.33 | 3.71E-06 | 0.00948 |
| 6 | Vldlr | 22359 | 1.224 | 6.325 | 10.30 | 3.80E-06 | 0.00948 |
| 7 | Abca8a | 217258 | 1.116 | 5.984 | 9.99 | 4.84E-06 | 0.00976 |
| 8 | Cnksr1 | 194231 | 1.246 | 3.741 | 9.71 | 6.07E-06 | 0.00976 |
| 9 | Sec14l2 | 67815 | −1.066 | 5.350 | −9.67 | 6.20E-06 | 0.00976 |
| 10 | 2610507I01Rik | 72203 | −4.421 | 3.956 | −9.56 | 6.88E-06 | 0.00976 |
| 11* | **Eps8l1** | 67425 | −4.100 | 2.602 | −9.47 | 7.41E-06 | 0.00976 |
| 12* | **Lama2** | 16773 | 1.475 | 6.078 | 9.37 | 8.05E-06 | 0.00976 |
| 13* | **Bnip3** | 12176 | 1.363 | 5.117 | 9.21 | 9.31E-06 | 0.00976 |
| 14 | Slc7a2 | 11988 | 1.437 | 5.507 | 9.19 | 9.46E-06 | 0.00976 |
| 15 | Pirb | 18733 | −1.242 | 6.794 | −9.15 | 9.78E-06 | 0.00976 |
| 16* | **Trim63** | 433766 | 2.476 | 7.151 | 9.02 | 1.09E-05 | 0.01019 |
| 17 | Cped1 | 214642 | 0.993 | 5.745 | 8.92 | 1.19E-05 | 0.01029 |
| 18 | Fam110c | 104943 | −3.115 | 0.935 | −8.87 | 1.25E-05 | 0.01029 |
| 19 | Klf9 | 16601 | 0.816 | 6.472 | 8.77 | 1.37E-05 | 0.01029 |
| 20* | **Klf1** | 16596 | −1.095 | 7.140 | −8.70 | 1.45E-05 | 0.01029 |
| 21 | Zim1 | 22776 | 1.597 | 1.624 | 8.55 | 1.66E-05 | 0.01029 |
| 22 | Abcg4 | 192663 | −0.884 | 5.372 | −8.49 | 1.76E-05 | 0.01029 |
| 23 | Tspo2 | 70026 | −1.010 | 5.864 | −8.46 | 1.80E-05 | 0.01029 |
| 24* | Unc5a | 107448 | −1.321 | 3.183 | −8.45 | 1.83E-05 | 0.01029 |
| 25* | **Peg3** | 18616 | 1.748 | 4.988 | 8.44 | 1.84E-05 | 0.01029 |
| 26 | Abca6 | 76184 | 1.576 | 3.620 | 8.44 | 1.84E-05 | 0.01029 |
| 27 | Slc15a2 | 57738 | 1.446 | 4.194 | 8.43 | 1.86E-05 | 0.01029 |
| 28 | Sycp1 | 20957 | −3.876 | −0.345 | −8.17 | 2.37E-05 | 0.01149 |
| 29 | Unc5cl | 76589 | −1.527 | 2.332 | −8.16 | 2.40E-05 | 0.01149 |
| 30 | Myo15 | 17910 | 1.291 | 2.933 | 8.09 | 2.56E-05 | 0.01149 |

The top 30 differentially expressed genes in marrow-flushed _Dmp1Cre.Efnb2_<sup>f/f</sup> femora from female 12-week-old mice compared to _Dmp1Cre_ controls. Autophagy-associated genes associated are in bold, with *
_Log 2 FC_ log 2 fold change, _Avg exprn_ average log 2 expression, _FDR_ false discovery rate

tested the effect of the autophagy inducer rapamycin on mineralizing Ocy454 cells. Indeed, Ocy454 mineralization was elevated by 24 h of rapamycin treatment (Fig. 5g). This provides direct evidence that stimulation of autophagy in osteocytes increases mineralization.

**More autophagosomes in _Dmp1Cre.Efnb2_<sup>f/f</sup> osteocytes.** We used transmission electron microscopy to assess bone-embedded osteocytes _in situ_ from adult _Dmp1Cre.Efnb2_<sup>f/f</sup> mice (Fig. 6). _Dmp1Cre.Efnb2_<sup>f/f</sup> osteocytes were grossly abnormal; their cell bodies were contracted, leaving a large open space within each lacuna (Fig. 6a–c). This was even seen by light microscopy in semi-thin sections, and when quantified throughout the cortical bone, there were significantly more osteocyte lacunae with small or absent osteocyte bodies (mean ± SEM _Dmp1Cre_: 32.6 ± 3.2, _Dmp1Cre.Efnb2_<sup>f/f</sup>: 49.6 ± 3.4; $p = 0.044$, by unpaired $t$ test, $n = 4$ mice; data from 104 to 127 lacunae counted per sample at eight sites, including both periosteal and endocortical regions).

Although cell bodies were small the _Dmp1Cre.Efnb2_<sup>f/f</sup> osteocytes were clearly viable and active (Fig. 6b, c), and exhibited extensive cell membrane ruffling, with abundant extrusions and matrix vesicles budding from the cell surface; the latter is consistent with EphrinB2-deficient osteocytes actively promoting mineralization in the surrounding matrix. Nuclei also had a higher proportion of condensed chromatin. Although the cell bodies within lacunae showed a striking shape change, dendritic processes appeared normally throughout the bone matrix inside the canaliculi (Fig. 6b, c), and a normal canalicular pattern was observed by silver staining (Supplementary Fig. 1). Within the cytoplasm, consistent with elevated LC3 detection in vitro,

_Dmp1Cre.Efnb2_<sup>f/f</sup> osteocytes exhibited abundant autophagosomes at various stages, more than observed in control samples (Fig. 7e–i). This included (1) condensations of degraded endoplasmic reticulum (ER), (2) partially formed autophagosomes (phagophores) actively encapsulating degraded ER cargo, (3) fully formed autophagosomes surrounded by a double membrane, including a single electron-dense region, and (4) lysosomes (Fig. 6i); this provides evidence that the type of autophagy promoted in EphrinB2-deficient osteocytes is ER-phagy. Sometimes, fully formed autophagosomes were observed within the matrix vesicles (Fig. 6h). These degraded ER and autophagosomes were not observed in control cells (Fig. 6a, d); the most similar structure seen in wild-type cells was an occasional rounded mitochondrion with clearly visible cristae and lacking the electron-dense region; usually the mitochondria were not spherical. This indicates high levels of both ER-phagy and matrix vesicle release in _Dmp1Cre.Efnb2_<sup>f/f</sup> osteocytes in situ.

**EphrinB2 increases osteocyte autophagosomes via RhoA-ROCK.** Since EphrinB2 has been reported to signal, at least in part, through RhoA GTPase/ROCK signalling[27], we next determined whether RhoA inhibition could mimic the effects of EphrinB2 deletion on mineralization. To test this, we used Kusa 4b10 cells, which differentiate into osteocytes[3], but mineralize at a lower level than Ocy454 cells. In these cells, mineralization was significantly elevated by two RhoA inhibitors, H1152 and Y27632 (Fig. 7a–d). We next investigated whether treatment with clustered EphrinB2-Fc would suppress autophagy, and whether this would be amelioriated by blocking RhoA-ROCK signalling. Neither H1152 nor EphrinB2-Fc, alone or in combination,

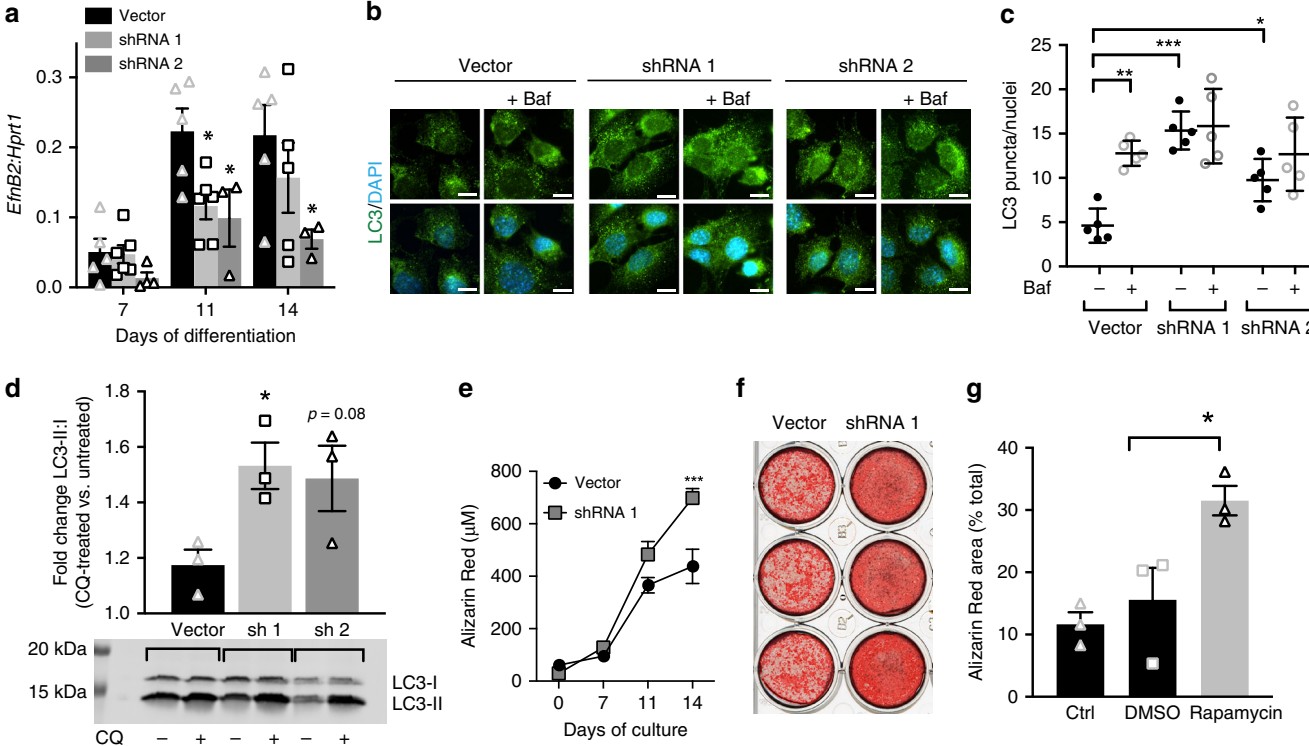

**Fig. 5** Confirmation of increased autophagy and mineralization in EphrinB2-deficient Ocy454 osteocytes, and effect of autophagy on mineralization. **a** *Efnb2* stable knockdown (short hairpin RNA (shRNA) 1 and shRNA 2) in Ocy454 cells differentiated for 7, 11, and 14 days, mean ± SD, 3–6 replicates, representative of three independent cultures; *$p < 0.05$ vs. vector by Student's $t$ test. **b**, **c** Autophagosomes (LC3 punctae) in *Efnb2*-deficient Ocy454 cells; bafilomycin (Baf) treatment (50 nM for 2 h) was used to block autophagosome degradation. Data shown are mean ± SD, five replicates, representative of three independent cultures; *$p < 0.05$; ***$p < 0.001$ as indicated by one-way analysis of variance (ANOVA). Scale bar = 10 μm **d** Fold change of LC3-II:I ratio in *Efnb2* shRNA knockdown cells treated with chloroquine (CQ) for 4 h compared to basal levels of each shRNA construct. Data are represented as mean ± SEM, three replicates, representative of three independent cultures, *$p < 0.05$ vs. vector control by one-way ANOVA. **e**, **f** Alizarin Red stain for mineralization in Ocy454 cells with vector or *Efnb2* shRNA 1 knockdown, grown under mineralizing conditions for 14 days. Data shown are mean ± SD, four replicates, representative of three independent cultures; ***$p < 0.001$ vs. vector by two-way ANOVA; **f** Alizarin Red-stained wells at day 14, showing three replicates. **g** Effect of rapamycin treatment (0.05 nM in dimethyl sulfoxide (DMSO) for 24 h) on Ocy454 cells grown under mineralizing conditions for 6 days. Data shown are mean ± SD, three replicates, representative of two independent cultures; *$p < 0.05$ vs. DMSO and $p < 0.01$ vs. untreated by one-way ANOVA

modified LC3 punctae number without bafilomycin (Fig. 7f). Bafilomycin treatment led to accumulation of LC3 punctae, and EphrinB2-Fc treatment significantly suppressed this (Fig. 7f), confirming less autophagosome formation. This was not observed when EphrinB2-Fc treatment was given with H1152, a RhoA-ROCK pathway inhibitor (Fig. 7f), indicating that EphrinB2-signalling inhibits autophagy through RhoA-ROCK signalling.

## Discussion

This work demonstrates that bone flexibility is maintained, and mineral accrual limited, by EphrinB2 signaling in osteocytes. In EphrinB2 deficiency, autophagy is increased, and although osteoid mineralization is initiated (MAR) at a normal rate, as soon as the process commences, secondary mineral deposition and maturation is accelerated, resulting in a brittle bone phenotype (Fig. 8). The control mechanisms for primary and secondary mineralization are therefore different, and osteocytes within the bone matrix contribute to secondary mineralization. This has major implications for understanding how osteocytes control bone strength.

Bone strength is determined by bone mass, and by bone matrix composition. Bone mass is determined by the balance in activity between bone-forming osteoblasts and bone-resorbing osteoclasts. Bone compositional strength is determined both by collagen content and orientation, and, within the collagen network, by mineral crystal content and nature. Defects in bone compositional strength have been noted to result from defective collagen deposition (as in osteogenesis imperfecta) or delayed mineralization initiation (as in osteomalacia or rickets). In this study, we report a third possible cause of bone fragility: accelerated mineralized bone matrix maturation (secondary mineralization). In brittle *Dmp1Cre.Efnb2f/f* bones, mineral:matrix ratio in the first region of mineral deposition was higher than controls. Since there was no difference in MAR (the rate at which osteoid mineralization is initiated), mineralization must proceed more rapidly after initiation: essentially, once mineralization starts, the mineral is deposited into the matrix at higher levels than normal. The normal gradual increase in mineral:matrix ratio, previously described in murine, rabbit, rat, and human cortical bone[13,14,28,29], was accelerated in the absence of EphrinB2.

This was not the only modification to mineral accumulation. Bone mineral crystals are a modified hydroxyapatite comprising calcium, phosphate, and hydroxyl ions, which can be replaced by fluoride, chloride, or carbonate. As bioapatite matures in the bone, carbonate:mineral ratio increases due to carbonate substitution for phosphate or hydroxyl ions[13,30]. *Dmp1Cre.Efnb2f/f* bones also showed a more rapid increase in carbonate incorporation within the bone matrix. Both mineral accrual and carbonate substitution within the mineral are accelerated in the absence of EphrinB2.

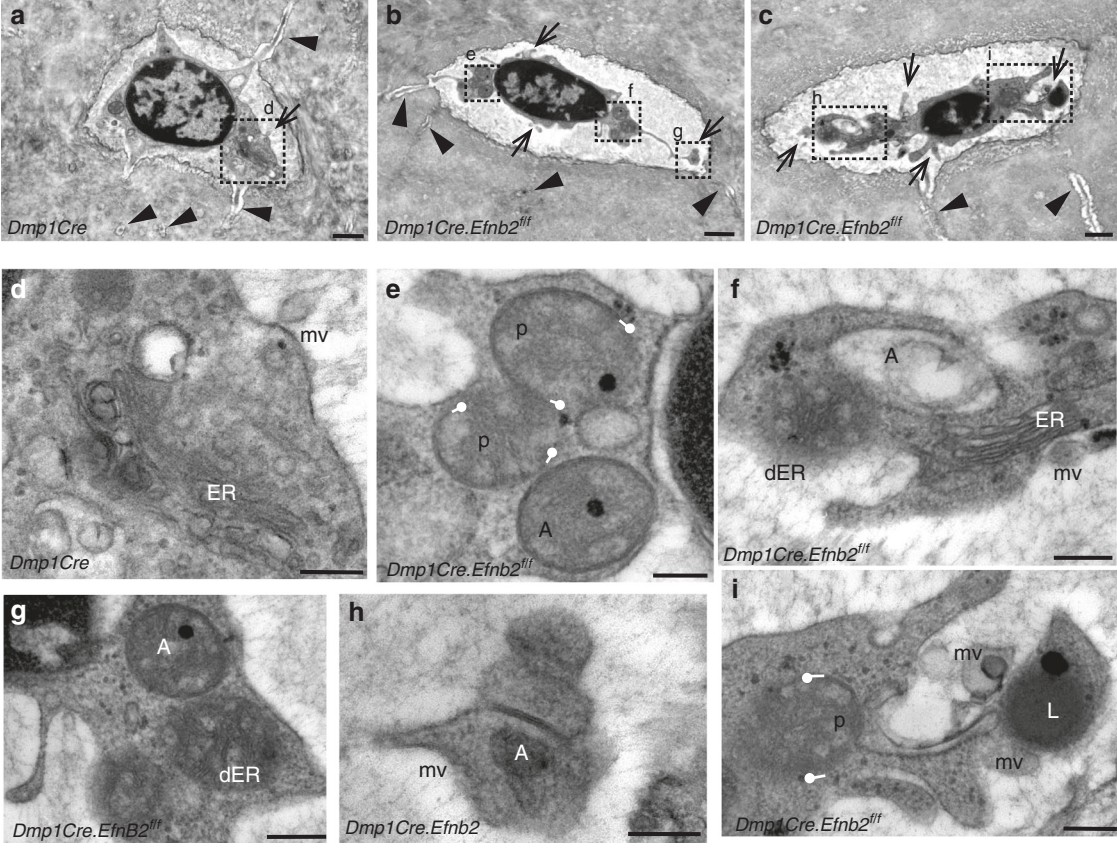

**Fig. 6** Transmission electron microscopy of bone-embedded osteocytes in *Dmp1Cre* (**a**, **d**) and *Dmp1Cre.Efnb2^{f/f}* (**b**, **c**, **e–i**) mice. **a–c** Low power images, showing typical osteocyte morphology in control *Dmp1Cre* bone (**a**) and in *Dmp1Cre.Efnb2^{f/f}* bone (**b**, **c**). *Dmp1Cre.Efnb2^{f/f}* osteocytes are grossly abnormal, showing contraction from the lacuna, extensive ruffling, and formation of matrix vesicles (arrows), but normal appearance of dendritic processes within the canaliculi (arrowheads); boxes show regions magnified in **d–i**. **d** Cytoplasm of a control *Dmp1Cre* osteocyte, showing a budding matrix vesicle (mv), and endoplasmic reticulum (ER) within the cytoplasm. **e–i** Features of *Dmp1Cre.Efnb2^{f/f}* osteocytes, including autophagomes (A) and lysosomes (L). Autophagosomes are either fully enclosed (**e–h**) or exhibit phagophores (p) in the process of encapsulating their cargo, which has the appearance of degraded ER (**e**, **f**, **i**); phagophores in the process of formation are delineated with white lollipops at the tips of the membrane with sticks facing the direction of the formed membrane. Functioning ER is also observed (**f**), along with unencapsulated degraded ER (dER), and many matrix vesicles (mv) in various stages of budding (**f–i**). Scale bars **a–c** = 1 μm, **d–i** = 250 nm

A third acceleration in bone matrix maturation was the reduction in the amide I:II ratio in *Dmp1Cre.Efnb2^{f/f}* bones. Amide I:II ratio represents peptide bond vibrations within the collagen molecule[31]. Amide I (C = O stretch) and amide II (C-N, N-H bend) molecular vibrations exhibit dichroism perpendicular and parallel to the collagen molecular triple helix axis, respectively. We have previously reported a decline in amide I:II ratio as mineral accumulates during normal bone maturation; this is likely due to increased compaction (or steric hindrance) in the perpendicular direction of the collagen molecule as mineral accumulates[13]. The earlier amide I:II ratio lowering in *Dmp1Cre. Efnb2^{f/f}* bone indicates more rapid collagen compaction as matrix mineral accumulates; essentially, the collagen is more compressed. Since the reduction in amide I:II was detected under the 0°, but not the 90°, polarizing filter, this may be specific to those collagen fibers aligned along the length of the bone, rather than those with radial orientation transverse to the bone. This suggests that the altered organization of longitudinal collagen fibers may be a main contributor to the EphrinB2-deficient phenotype, and these fibers may be more important for absorbing mechanical forces along the tibial length.

Another indicator of accelerated bone matrix maturation is the greater osteocyte lacunar density in *Dmp1Cre.Efnb2^{f/f}* bone. As osteoid is deposited, some osteoblasts are incorporated into the bone matrix and differentiate into osteocytes. Since osteoid deposition is not accelerated, and mineral appositional rate (initiation of mineralization) is unchanged in *Dmp1Cre.Efnb2^{f/f}* bone, the increased osteocyte density may indicate more rapid osteocyte incorporation into the bone matrix. *Dmp1Cre* targets late osteoblasts at the stage where they become embedded within the newly formed osteoid;[32] the greater density of osteocytes incorporated may promote mineral accumulation and carbonate incorporation in *Dmp1Cre.Efnb2^{f/f}* bones, but whether this is secondary to the increased mineralization or is causative is not known and would be difficult to test.

Although this is the first report of a low amide I:II ratio being associated with bone fragility, the cause of the brittle phenotype is likely to be the combination of low amide I:II, and high mineral: matrix and carbonate:mineral. An association of high carbonate with increased bone fragility is consistent with a higher carbonate: mineral ratio in bone specimens from women with post-menopausal osteoporosis[33] and with greater fracture suscept-ibility[34]. High mineral:matrix ratio has also been reported in other brittle bone conditions, such as patients with atypical femoral fracture[35], and murine osteogenesis imperfecta models[36]. In the latter case, it is accompanied by defects in collagen or cartilage content and in osteoblast function, but the present model exhibited no change in osteoblast function.

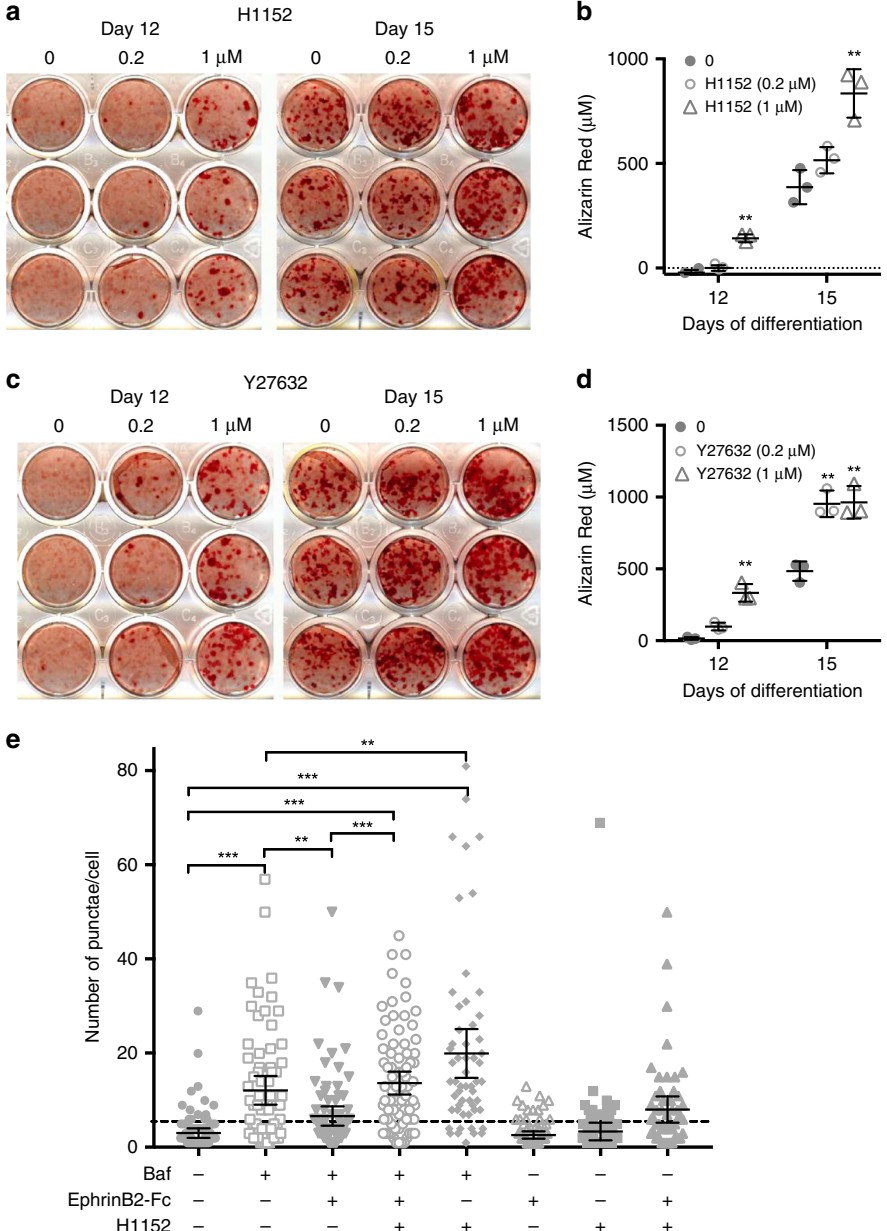

**Fig. 7** RhoA-ROCK inhibition increases mineralization and EphrinB2 inhibits autophagy through a RhoA-ROCK-dependent mechanism. **a–d** Mineralization by Kusa 4b10 cells is enhanced when RhoA-ROCK is inhibited with either H1152 or Y27632. Kusa 4b10 osteocytes were cultured under mineralizing conditions and treated with H1152 (**a**, **b**) or Y27632 (**c**, **d**) at the doses indicated for the duration of the culture. Alizarin Red staining and quantitation is shown. Data are mean ± SD of three replicates from a representative of three independent experiments; **$p < 0.01$ vs. untreated (0) at same day of differentiation by two-way analysis of variance (ANOVA). **e** Autophagy is suppressed by EphrinB2 in a RhoA-ROCK-dependent manner. LC3 punctae formation in Ocy454 cells, either untreated or treated with bafilomycin A1 (Baf) (50 nM), in the presence and absence of clustered EphrinB2-Fc (1 μg/mL) and H1152 (50 μM) for 1 h. Punctae number per cell from two independent experiments, each performed in duplicate, with 95% confidence interval (CI); ***$p < 0.0001$; **$p < 0.001$ for comparisons shown, by one-way ANOVA with correction for multiple comparisons

The processes controlling mineral accumulation in the bone are poorly defined and our data suggest that autophagic processes in the osteocyte contribute. Unbiased RNA-sequencing detected no changes in mRNA for genes known to regulate mineralization. Instead, it identified a number of dysregulated autophagy-associated genes. This led us to further investigations showing elevated autophagosome numbers both in the EphrinB2 knock-down osteocyte cell line, and in *Dmp1Cre.Efnb2*$^{f/f}$ osteocytes.

Autophagy is a group of lysosome-based recycling and secretory processes contributing to diverse cellular functions, such as adaptation to starvation, quality control of intracellular processes,

protein secretion, and elimination of intracellular microbes[37]. Autophagy can be broadly categorized into two classes, canonical degradative macro-autophagy, which involves *Atg* proteins, and micro-autophagy (including mitophagy and ER-phagy), which can be *Atg*-independent. While exploration of autophagy in osteocytes is in its infancy, previous work demonstrated that autophagy increases during osteoblast differentiation[38], and mice with osteoblast lineage or osteocytic deletion of either *Atg5* or *Atg7* have reduced autophagy and osteoblast numbers[38–40]. In our *Dmp1Cre.Efnb2*$^{f/f}$ mice osteoblast numbers were not altered, indicating that the autophagic processes modified in

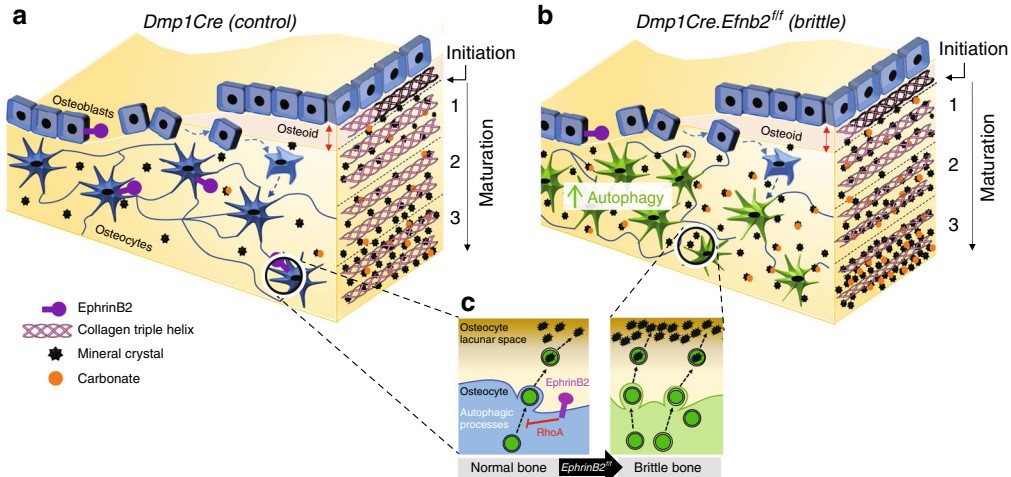

**Fig. 8** Model of osteocytic EphrinB2 regulation of bone matrix composition. **a** In control bone (*Dmp1Cre*) both osteoblasts and osteocytes express EphrinB2. Osteoblasts reside on the bone surface, and pass through a transition (dashed arrows) to become mature, matrix-embedded osteocytes. Osteoblasts deposit collagen-containing osteoid (triple helical collagen fibers, shown on the right). In region 1 of newly formed bone, mineral deposition is initiated. Mineral crystals (black stars) and carbonate (orange circles) continue to accumulate, and collagen fibers become more compact as the matrix matures in regions 2 and 3. **b** *Dmp1Cre.Efnb2^{f/f}* mice express EphrinB2 in osteoblasts, but not osteocytes. *Dmp1Cre.Efnb2^{f/f}* osteocytes have increased autophagy (green cells). Osteoid deposition occurs normally, and the initiation of mineralization commences at the same rate, leading to osteoid of the same thickness (red double-headed arrow). As soon as mineralization starts, mineral is deposited in *Dmp1Cre.Efnb2^{f/f}* bone at a greater level than control in regions 1, 2, and 3 to reach a level with more mineral, more carbonate substitution, and more compact collagen fibers than controls. Ultimately this leads to more brittle bone. **c** Close-up of the cell membrane in osteocytes in control and *Dmp1Cre.Efnb2^{f/f}* bone: we propose that, in control mice, EphrinB2 suppresses autophagic processes and limits matrix vesicle release via RhoA-ROCK signalling. In *Dmp1Cre.Efnb2^{f/f}* mice loss of this inhibition leads to a high level of matrix vesicle release, elevated mineralization, and a brittle bone matrix

EphrinB2-deficient osteocytes do not change their signals that control osteoblast differentiation. Furthermore mRNA levels of *Atg* family members were not changed in *Dmp1Cre.Efnb2^{f/f}* bones, suggesting that EphrinB2 deficiency does not modify canonical degradative macro-autophagy. Indeed, the top genes dysregulated in the EphrinB2-deficient bones have been associated specifically with a subset of autophagic processes, including mitophagy (degradation of mitochondria) and ER-phagy (microautophagic degradation of the ER), as follows: *Fam134b*[16], *Fbxo32*[17], *Lama2*[18], *Bnip3*[19], *Trim63*[20], *Peg3*[21], *Eps8l1*[22], *Klf1*[23], *Tspo2*[24], and *Unc5a*[25]. The abundant autophagosomes containing degraded ER in osteocytes of *Dmp1Cre.Efnb2^{f/f}* mice is consistent with up-regulated ER-phagy, an *Atg*-independent process[41]. When we used rapamycin as a broad-acting stimulus of autophagy, mineralization by osteocytes was increased, providing proof of principle that stimulating autophagy can stimulate mineralization by osteocytes. Whether rapamycin treatment in vivo would increase bone stiffness by stimulating mineralization is not known, and effects on other cell types are likely to mask such action. We recently tested rapamycin treatment in wild type mice and in the α2(I)-G610C model of osteogenesis imperfecta[42]. Although rapamycin increased trabecular bone mass, such treatment also reduced cortical bone growth, thereby reducing bone strength[42]. An increase in mineralization of bone deposited during the rapamycin treatment period may have also played a part in weakening the bone structure. Whether ER-phagy is directly involved in mineral release remains to be tested, and will require the development of ER-phagy-specific modulators.

The process of mineral incorporation within the extracellular matrix is not well understood, and remains highly controversial. The process involves matrix vesicles, which are extracellular membrane-enclosed vesicles released by budding from cells at the site of mineralization[43], as seen in the *Dmp1Cre.Efnb2^{f/f}* osteocytes. Matrix vesicles were originally reported to lack mineral, and to accumulate poorly crystalline mineral only after their expulsion

from the cell and immobilization in the collagen matrix[43]. The amorphous mineral formed is subsequently nucleated (to ordered crystals) in a process driven by contact with collagen, and by secreted nucleators[44,45]. In contrast, a recent in vitro study reported calcium phosphate crystals within intracellular vesicles of osteoblast-like cells, and suggested that these are released as mineral-containing matrix vesicles[46]. Nollet et al.[38] also suggested that osteoblasts could use autophagosomes as vehicles for apatite crystal secretion into the extracellular space via exocytosis (i.e., secretory autophagy), and observed minerals within autophagosomes in cultured osteosarcoma cells[38]. Whether osteocyte-derived matrix vesicles that mediate secondary mineralization are distinct from those released by osteoblasts is not known, nor is it known yet whether matrix vesicles from EphrinB2-deficient osteocytes differ from wild type in their mineral content or interaction with the collagen matrix. This would require study of cultured osteocytes within a mineralized collagen substrate, or assessment of mineral content within autophagosomes in bone samples. Methods for culturing osteocytes within a mineralized collagen substrate are not yet developed, and due to the high mineral content of adult murine bone, decalcification was required for TEM imaging of the bone specimens we studied; this question therefore remains unresolved.

EphrinB2 has been previously reported to activate RhoA-ROCK signalling in melanoma cells[27]. Our data indicate that EphrinB2 suppresses autophagy in osteocytes, at least in part, through a RhoA-ROCK-dependent pathway. Since RhoA-ROCK inhibition both prevented the effects of EphrinB2-Fc on autophagy and stimulated mineralization, this is likely to be at least one pathway through which EphrinB2 limits mineralization and autophagy. Whether RhoA-ROCK inhibition in osteocytes would stimulate mineralization in vivo is not known; investigating this in future would require a targeted approach, since RhoA-ROCK signalling stimulates osteoblast commitment[47] and is required for osteoblast survival[48].

A comparison of this mouse model with our earlier model lacking EphrinB2 throughout the osteoblast lineage provides new information about the specific stages of osteoblast/osteocyte differentiation and their roles in bone mineralization. That model (*OsxCre.Efnb2f/f*) did not exhibit brittle bone, but had more elastic bones because primary mineralization initiation was delayed[11]. This indicated that the stage of osteoblast differentiation at which primary mineralization is initiated is beyond the EphrinB2:EphB4 osteoblast differentiation checkpoint. In *Dmp1Cre.Efnb2f/f* mice, osteoblasts survive past the checkpoint at which anti-apoptotic EphrinB2 action is required; this allowed us to define an action for EphrinB2 later in osteoblast differentiation. Normal initiation of mineralization in *Dmp1Cre.Efnb2f/f* bone indicates that the stage of osteoblast differentiation controlling this process is not only after the EphrinB2:EphB4 checkpoint but also before the stage of *Dmp1Cre* expression. In contrast, after *Dmp1Cre* is expressed, osteocytes do not control mineralization initiation, but control the rate at which minerals accumulate (Fig. 8). Since exogenous PTH and endogenous PTHrP promote EphrinB2 expression in both osteoblasts[9] and osteocytes, they may both promote initiation of osteoid mineralization and restrain mineral accrual. While there is no evidence for altered mineral accrual with PTH pharmacological administration[13], the reduced material strength of mice with osteocyte-specific PTHrP deletion[49] suggests that PTHrP may regulate physiological mineral accrual through EphrinB2-dependent actions to suppress autophagy in the osteocyte.

These findings have implications for other cell types and organ systems. EphrinB2 in odontoblasts, which are closely related cells to osteoblasts and osteocytes[50], may control mineralization in teeth. EphrinB2 in hypertrophic growth plate chondrocytes may control cartilage mineralization during endochondral ossification; this may contribute to the cartilage accumulation in neonate *Osx1Cre.Efnb2f/f* mice[51]. Pathological mineralization, such as that in heterotopic ossification after trauma[52], and vascular or renal calcification[53] may also be limited by EphrinB2, since it is expressed in muscle, blood vessels, and kidney.

Sex differences in bone structure, strength, and remodeling exist in all mammalian species, including humans and mice[54,55], and it is common for genetically altered mice to exhibit sex-specific phenotypes[56–58]. Both the strength defect and the highly mineralized bone of *Dmp1Cre.Efnb2f/f* mice detected here was only observed in female mice. Our earlier work using *Osx1Cre* to initiate EphrinB2 deletion earlier in the osteoblast lineage also resulted in a female-specific phenotype[11]. However, there is no evidence for direct regulation of *Efnb2* in Ocy454 cells by either estradiol or testosterone. We suggest that changes in bone matrix composition may play a more significant role in determining bone strength in female bones because they have higher mineral content and more rapid bone remodeling than male bones[58,59].

In conclusion, EphrinB2 is required to limit autophagosome numbers in osteocytes, and to prevent formation of brittle bone. Osteocytic EphrinB2 limits autophagy through RhoA, and this may be responsible for limiting mineral accumulation and carbonate substitution within the bioapatite matrix and restraining collagen fiber compaction (Fig. 8). The EphrinB2-deficient mouse model indicates that such a mechanism can regulate bone strength independently of both bone size and osteoblast and osteoclast activities, and implies that autophagic processes within the osteocytes may modify bone quality by regulating mineral secretion.

## Methods

**Cell culture**. Ocy454 cells were cultured in α-modified Eagle's medium (α-MEM) supplemented with 10% fetal bovine serum (FBS) and 1% penicillin–streptomycin–amphotericin B (PSA) and Glutamax[60]. Cells were maintained in permissive conditions (33 °C) and differentiated to osteocytes at 37 °C[60]. Ocy454 cells were plated at $2.5 \times 10^5$ cells per well in six-well plates. Cells were grown at the permissive temperature (33 °C) for 3 days prior to transferring to 37 °C for differentiation. To study *Efnb2* mRNA regulation by PTH and PTHrP, Ocy454 cells at day 14 of differentiation were serum depleted overnight in α-MEM supplemented with 1% FBS and 1% PSA and Glutamax. Cells were then treated with human PTH(1–34) (Bachem) (10 nM) or human PTHrP(1–141)[61] (10 nM) for 6 h. After 6 h, cells were washed with phosphate-buffered saline (PBS) and RNA samples were collected as described below. Undifferentiated Ocy454 cells were infected with *Pthlh* knockdown virus[6], selected with puromycin (5 µg/mL), and then cultured at permissive temperature (33 °C) before transfer to 37 °C for differentiation. Cells were assessed by quantitative real-time PCR (qRT-PCR) at day 0, 7, and 14[6]. Rapamycin (0.05 nM, Sigma) was administered to Ocy454 cells for 24 h after 6 days of differentiation in mineralizing media such that mineralization had commenced at the time of treatment. Dimethyl sulfoxide at the same volume was used as a negative control. Cells were stained with Alizarin Red (Sigma). Since folding of the cell layer occurred at the edge of the wells and trapped Alizarin Red stain, mineralization was quantified using areal thresholding in a circular region (avoiding the folded areas) with ImageJ; values are expressed as the percentage of the area measured[62].

Kusa 4b10 cells, a murine stromal cell line, capable of osteocyte differentiation[63], were differentiated in mineralizing media (α-MEM with 15% heat-inactivated FBS, 50 µg/mL ascorbate, and 5mM β-glycerophosphate), and then treated with H1152 or Y27362 (0.2 and 1 µM) with each media change until days 12 or 15, at which point cells were fixed and stained with Alizarin Red (Sigma), which was solubilized and quantified against a standard curve[9].

**qRT-PCR analysis**. RNA was extracted from cultured cells by RNA extraction kits with On-Column DNase digestion (Qiagen, Limburg, The Netherlands) or TriSure reagent (Bioline, London, UK). Extracted RNA was DNase treated with Ambion TURBO DNA-free Kit (Life Technologies) and quantified on a NanoDrop ND1000 Spectrophotometer (Thermo Scientific, Wilmington, DE, USA). Complementary DNA (cDNA) was synthesized from total RNA with AffinityScript cDNA Synthesis Kits (Agilent Technologies, Santa Clara, CA, USA). Gene expression for PTH- and PTHrP-treated Ocy454 cells were quantified on a Stratagene Mx3000P QPCR System (Agilent) with SYBR Select Master Mix (Applied Biosystems) with primers specific to *Efnb2*: forward 5′-GTGCCAGACAAGAGCCATGAA-3′ and reverse 5′′-GGTGCTAGAACCTGGATTTGG-3′[9]. Gene expression for *Pthlh* knockdown cells was analyzed using the Multiplex SensiMix II Probe Kits (Bioline, London, UK) with primers specific to *Efnb2*[10]. Gene expression levels were normalized to hypoxanthine phosphoribosyltransferase 1 (*Hprt1*) expression[10]. Relative expression was quantified using the comparative CT method ($2^{-(Gene\ Ct\ -\ Normalizer\ Ct)}$).

**Mice**. Dmp1Cre mice (Tg(Dmp1Cre)[1]Jqfe) (using the DMP1 10-kb promoter region) were obtained from Lynda Bonewald (University of Kansas, Kansas City, KS, USA)[32] and EphrinB2-floxed (Efnb2[tm1And]) mice were obtained from David J. Anderson (Howard Hughes Medical Institute, California Institute of Technology, Pasadena, CA, USA);[64] all were backcrossed onto C57BL/6 background. Mice hemizygous for Dmp1Cre were crossed with Efnb2f/f mice to generate Dmp1Cre. Efnb2f/w breeders, which were used to generate Dmp1Cre.Efnb2f/f mice and Dmp1Cre.Efnb2w/w (Dmp1Cre) littermates or cousins; the latter were used as controls for all experiments. Bone samples were collected at 6 and 12 weeks of age. Calcein (20 mg/kg) was administered by intraperitoneal injection 7 and 2 days before, and mice were fasted for 12 h prior to collection of tibiae and femora from male and female mice; sample numbers were chosen based on previous studies using adult genetically altered mice with alterations in bone mass and strength[56]. All animal procedures were conducted with approval from the St. Vincent's Health Melbourne Animal Ethics Committee. All animals were assigned numbers at the time of birth, and all animal procedures were conducted without the knowledge of the genotype of the mice (blinded). At tissue collection, all samples were allocated a second non-identifying number and all further analyses were conducted in a blinded fashion and in random order, but no formal randomization procedure was used.

**Confirmation of *Efnb2* mRNA targeting in osteocytes**. To confirm specific targeting of *Efnb2* mRNA in osteocytes, *Dmp1Cre.Efnb2f/f* mice were crossed with (Tg(Dmp1-Topaz)[11]kal) mice obtained from Dr. Ivo Kalajzic, University of Connecticut Health Science Center[65] to allow GFP labelling of osteocytes and their purification by FACS[66]. Osteocytes from 6-week-old *Dmp1Cre.Dmp1-GFP-Tg. Efnb2w/w* and *Dmp1Cre.Dmp1-GFP-Tg.Efnb2f/f* mice were isolated from marrow-flushed long bones by seven sequential 15-min digestions in 2 mg/mL dispase (Gibco, Grand Island, NY, USA) and 1 mg/mL collagenase type II (Worthington, Lakewood, NJ, USA). Fractions 2–7 were collected, pooled, and resuspended in α-MEM (Gibco, Grand Island, NY, USA) containing 10% FBS and centrifuged at 400 × *g* for 5 min. Pellets were resuspended in FACS buffer before cell sorting. Prior to sorting, dead cells and debris were removed based on side scatter (SSC) area and forward scatter (FSC) area, and doublets were excluded based both on SSC width (W) vs. SSC height (H) and on FSC-W vs. FSC-H. Cells were sorted with excitation 488 nm and 530/30- or 530/40-emission filter for GFP on a BD FACS Influx cell

sorter (BD Biosciences, Scoresby, Australia). RNA was extracted using Isolate II Micro RNA Kit (Bioline, London, UK). cDNA was prepared using a Superscript III Kit (Thermo Fisher, Scoresby, Australia). Gene expression levels of *Efnb2* were measured using primers directed to the genetically targeted region[11] normalized to β2 microglobulin (*B2m*) and hydroxymethylbilane synthase (*Hmbs*) or to *Hprt1* using primers listed in Supplementary Table 7.

**Three-point bending and RPI.** Structural and material strength were analyzed in femora from male and female 12-week-old mice by three-point bending[56]. Load was applied in the anterior–posterior direction of the femoral mid-shaft between two supports that were 6.0 mm apart. Load–displacement curves were recorded at a crosshead speed of 1.0 mm/s using an Instron 5564A dual column material testing system, and Bluehill 2 software (Instron, Norwood, MA, USA). Ultimate force, ultimate deformation, post-yield deformation, and energy absorbed to failure were measured from the load–displacement curves. Cortical dimensions including anteroposterior (AP) and mediolateral (ML) widths were measured at the cortical mid-shaft by micro-computed tomography (microCT). Moment of inertia was calculated based on AP, ML, and cortical thickness[6]. The material properties, corrected for geometry, were calculated for each bone to obtain ultimate stress, ultimate strain, and toughness[56].

Local bone material properties were examined by RPI at the femoral mid-shaft with a BP2 probe assembly apparatus (Biodent Hfc; Active Life Scientific Inc., Santa Barbara, CA, USA)[56]. A 2 N maximum indentation force was achieved by manually applying a 300 g reference force to femora. Five measurements were taken per sample using a 2 N, 10-cycle indentation protocol. Pre- and post-experiment measurements were taken on a polymerized methyl methacrylate (MMA) block to ensure that probe assembly was not affected during testing. Indentation distance increase was measured as the indentation distance in the last cycle relative to the first cycle.

**Hydroxyproline assay in hydrolyzed bone samples.** Following mechanical testing, all fragments of the broken femora were flushed of marrow, hydrolyzed, and used for hydroxyproline assay to measure the collagen content[31]. Femoral samples were cut with scissors at the mid-diaphysis and the distal half of the femur was flushed of marrow. The bone was weighed before and after dehydration in a 37 °C incubator for 1 h. Each sample was hydrolyzed by incubation overnight at 120 °C in 1 mL of 5 M HCl per 20mg of bone. Fifty microliters of each sample was used, and six serial dilutions of the hydroxyproline standard (1 mg/mL) were used to generate a standard curve. Fifty microliters of chloramine T (Sigma-Aldrich) was added to each reaction and incubated at room temperature for 25 min. Five hundred microliters of Ehrlich's Reagent (dimethyl-amino-benzaldehyde (Sigma-Aldrich), *n*-propanol (99%, Sigma-Aldrich), perchloric acid (70%, AnalAR)) was added to each reaction and incubated at 65 °C for at least 10 min. One hundred microliters of each reaction was pipetted into a 96-well plate and absorbance was measured using a POLARstar plate reader at 550 nm and interpolated on the standard curve[67].

**Histomorphometry.** Tibiae and femora from 12-week-old female mice were fixed in 4% paraformaldehyde, and embedded in MMA. To image the osteocyte network, contralateral tibiae were decalcified in EDTA, embedded in paraffin, sectioned, and stained with Ploton silver stain[10]. Histomorphometry was conducted on MMA-embedded samples with periosteal parameters measured on the tibial medial mid-shaft (1500 μm from the base of the growth plate)[68] and trabecular parameters were measured in the secondary spongiosa, commencing 370 μm below the growth plate, in a 1110 μm² region in the proximal tibia[68] (Osteomeasure; Osteometrics, Atlanta, GA, USA).

**Backscatter electron microscopy.** Polarized light microscopy and backscatter electron microscopy (BSEM) were performed on 100-μm-thick transverse sections from the methacrylate-embedded femoral mid-shaft cut with a water-cooled Iso-met Saw (Buehler, Lake Bluff, IL, USA). Irregular fiber orientation was measured as woven bone, and well-aligned fiber orientation was measured as lamellar bone using the Osteomeasure system (Osteometrics, Decatur, GA, USA)[56]. Gray-level images of mineralized cortical bone were acquired by BSEM to identify osteocyte lacunae on both anterior and posterior sides of the femur at ×500 magnification (~600 μm in length) using an FEI Quanta FEG 200 solid-state backscattered scanning electron microscope. Imaging was performed at low vacuum using water at a 9.8 mm working distance. Lacunae were quantified using MetaMorph (v.7.8.3.0; Molecular Devices, Sunnyvale, CA, USA) by establishing an inclusive threshold for dark objects to distinguish bone from background. Integrated morphometry analysis (IMA) was applied with a 2.43–24.31 μm² filter for lacunar area, to exclude cracks and blood vessels. Thresholded bone area, total lacunar area, osteocyte lacunar size, size of the largest 20% of osteocyte lacunae, and osteocyte lacunar density were measured. The osteocyte lacuno-canalicular network was examined by Ploton silver staining[69] of paraffin-embedded tibiae.

**Micro-computed tomography.** MicroCT was performed on femora using the SkyScan 1076 System (Bruker-microCT, Kontich, Belgium). Images were acquired using the following settings: 9 μm voxel resolution, 0.5 mm aluminum filter, 50 kV

voltage, and 100 μA current, 2600 ms exposure time, rotation 0.5°, frame averaging = 1. Images were reconstructed and analyzed using NRecon (version 1.6.9.8), Dataviewer (version 1.4.4), and CT Analyzer (version 1.11.8.0). The femoral trabecular analysis region of interest (ROI) was determined by identifying the distal end of the femur and calculating 10% of the total femur length towards the mid-shaft, where an ROI of 15% of the total femur length was analyzed. Bone structure was measured using adaptive thresholding (mean of min and max values) in CT analyzer. Cortical analyses were performed in a region of 15% of the femoral length commencing from 30% proximal to the distal end of the femur and extending toward the femoral mid-shaft. The lower thresholds used for trabecular and cortical analysis were equivalent to 0.20 and 0.642 g/mm³ calcium hydroxyapatite (CaHA), respectively. Ct.TMD was analyzed in the same cortical ROI, as described above. TMD calibration was performed using two phantom rods with concentrations of CaHA of 0.25 and 0.75 g/cm³, scanned under the same conditions and settings as the samples.

**Synchrotron FTIR microspectroscopy.** sFTIRM was used to examine bone composition at the cortical diaphysis in 3 μm longitudinal sections of MMA-embedded tibiae. Sections were imaged using a Bruker Hyperion 2000 IR microscope coupled to a V80v FTIR spectrometer located at the IR Microspectroscopy beamline at the Australian Synchrotron[13]. Sections were placed on 22 mm diameter × 0.5 mm polished barium fluoride (BaF₂) windows (Crystan Limited, UK). The microscope video camera was used to image the cortical diaphysis (1.5 mm from the base of the growth plate), at the same location used for histomorphometric measurements of periosteal mineral apposition (Fig. 5a). This location is ideal for assessing mineral apposition on a formation surface, without any prior remodeling because mouse bone lacks intracortical Haversian systems. sFTIRM mapping was performed with the synchrotron source, with a 15 × 15 μm aperture. Spectra were collected from three regions progressing perpendicularly into the cortex, with the first positioned at the periosteal edge (Fig. 5a). Spectra were collected in the mid-infrared (IR) region from 750 to 3850 cm⁻¹ using a narrowband mercury cadmium telluride detector, at 8 cm⁻¹ spectral resolution and 128 co-added scans per pixel spectral resolution in transmission mode. A matching background spectrum was collected through clear BaF2. For each sample, MMA reference spectra were collected within the embedding material. All data acquisition was undertaken with Bruker OPUS version 6.5 and data analysis used OPUS version 7.2.

After acquisition, raw spectra for each region and sample were baseline corrected using a three-point baseline at 1800, 1200, and 800 cm⁻¹. Residual MMA absorbance peaks were then subtracted using the relevant MMA reference spectrum for each sample by iterative manual subtraction. A residual 1730 cm⁻¹ MMA band remained after MMA subtraction, which was not used for analysis. Spectroscopic parameters calculated were integrated peaks areas of the following bands: phosphate (1180–916 cm⁻¹), amide I (1588–1712 cm⁻¹) and II (1600–1500 cm⁻¹), and carbonate (890–852 cm⁻¹). Ratios were calculated as follows: mineral:matrix ratio (1180–916 cm⁻¹/588–1712 cm⁻¹), carbonate:phosphate ratio (1180–916 cm⁻¹/890–852 cm⁻¹) ratio and amide I:II ratio (1588–1712 cm⁻¹/1600–1500 cm⁻¹)[70]. Collagen crosslinking was determined by spectral curve fitting of the amide I and amide II peaks using Grams/AI (Version 9.2, Thermo Scientific, USA). The second derivative of each peak was used to estimate subpeak positions at ~1660 and 1690 cm⁻¹.

**Polarized light FTIR imaging.** To assess collagen compaction by a second method, pFTIRI was applied to the same 3 μm tibial sections at the analysis region used for sFTIRM. Larger regions were imaged using a 340 × 340 μm aperture on either side of the 1500 μm mid-point. Tissue sections were scanned using a Hyperion 3000 spectral imaging system equipped with a Vertex-70 spectrometer (Bruker, Germany), a liquid-N2-cooled focal plane array (FPA: 128 × 128 elements; 40 ×4 0μm each) detector and a Globar source. The FPA detector was continuously maintained at liquid-N2 temperature by an automated refilling system (Norhof LN2 cooling system #606; Maarssen, The Netherlands). The microscope and spectrometer were also continuously N2 purged and an insulation box protected the sample stage from ambient air. For all FTIR image acquisitions, a ×15 magnification level and condenser were used. High-resolution FTIR images were obtained for microscopic analysis of tissue sections; 200 scans and an 8 cm⁻¹ spectral resolution were used for image acquisition (140 ms FPA detector exposure per scan; spectral range = 3800–900 cm⁻¹). All FTIR images had an individual pixel dimension of 2.6 × 2.6 μm, thus at ~λ/2 for the mid-IR spectral interval. All IR images were obtained in transmission mode. The images were obtained from subroutines of the Opus 7.5 software (Bruker-Optics, France). To quantify region-specific changes in amide I:II, the diaphyseal cortex was divided into three equal parts: periosteal, central, and endosteal ROIs. Average spectra were extracted from each region, baselined and MMA-subtracted, as above, but using CytoSpec 1.4.0.3 (Bruker). Integrations for the amide I and II peaks were: amide I (1590–1730 cm⁻¹) and amide II (921–1190 cm⁻¹). Since collagen is by far the major protein component of bone, we interpreted the amide I:II ratio as peptide bond vibrations of the collagen molecule. The amide I (C = O stretch) and amide II (C-N, N-H bend) molecular vibrations exhibit dichroism perpendicular and parallel to the collagen molecular triple helix axis, respectively[15,31]. The FTIR microscope was coupled with two polarizing filters to measure molecular

orientation of collagen fibers relative to the plane of the tissue section. The 0° polarizing filter was used to measure bonds in plane (parallel to the section). The 90° polarizing filter was used to measure bonds out of plane (perpendicular to the section). This allowed quantification of collagen fibers aligned in different directions through the diaphyseal cortex.

**RNA-sequencing**. RNA samples were collected from flushed femora of 12-week-old female Dmp1Cre.Efnb2$^{f/f}$ and cousin-bred control mice. Samples were collected on two occasions: on day 1, two Dmp1Cre.Efnb2$^{f/f}$ and three control samples were collected, while on day 2, one sample of each genotype was collected. Bones were snap frozen in liquid nitrogen, and then homogenized in QIAzol lysis reagent with a Polytron PTA 20S homogenizer at 4 °C prior to RNA extraction with a RNeasy Lipid Minikit (Qiagen). RNA-sequencing was conducted on an Illumina HiSeq at the Australian Genome Research Facility to produce 100 bp paired-end reads. Reads were mapped to the mouse genome (mm10) using Rsubread[71]. Read counts were obtained using featureCounts and Rsubread's inbuilt mm10 annotation[72]. Gene annotation was obtained from the NCBI gene information file (downloaded 4 October 2016). Statistical analysis used the limma software package[73]. Genes were retained in the analysis if they achieved at least 0.65 read counts per million (cpm) in at least three samples. Immunoglobulin gene segments, ribosomal genes, predicted and pseudo genes, sex-linked genes (Y chromosome and Xist), and obsolete Entrez Gene IDs were filtered out. Quantile normalization was applied and read counts were transformed to log 2 cpm. Linear models were used to test for expression differences between deficient vs. control samples. The day of sample collection was included in the linear model as a blocking factor. Empirical sample quality weights were estimated[74]. Differential expression between the genotypes was assessed using empirical Bayes moderated t statistics allowing for an abundance trend in the standard errors and for robust estimation of the Bayesian hyperparameters[75]. The Benjamini and Hochberg method was used to adjust the p values so as to control the false discovery rate.

**Knockdown of EphrinB2 in Ocy454 cells**. Two shRNA constructs were used to knock down EphrinB2: shRNA 1 (5′-CGG-GTG-TTA-CAG-TAG-CCT-TAT-3′) and shRNA 2 (5′-CAG-ATT-GTG-TAC-ATA-GAG-CAA-T-3′), both obtained from Sigma-Aldrich (St. Louis, MO, USA). shRNA were cloned into the PLKO lentiviral vector[6] and infected by retrovirus into undifferentiated Ocy454 cells. Infected cells were selected with puromycin (5 μg/mL) and cultured at permissive temperature (33 °C) before transfer to 37 °C for differentiation. Knockdown was validated by qRT-PCR using multiplex primers, as above.

**Analysis of autophagy and mineralization in Ocy454 cells**. To assess LC3 puncta formation, undifferentiated Ocy454 (vector and EphrinB2-deficient) cells were grown on 22 × 22 mm$^2$ glass coverslips in a 24-well plate as above with Dulbecco's modified Eagle's medium (DMEM). When cells reached 70% confluency, they were treated for 2 h with bafilomycin A1 (50 nM, Enzo, BML-CM110), or remained in media. Following this cells were washed twice in PBS, fixed with 3.7% paraformaldehyde, and then quenched with DMEM/HEPES (pH 7.0). They were washed in PBS and then permeabilized with 0.2% NP-40 for 30 min. Later the cells were washed with PBS/1% bovine serum albumin (BSA) twice and then left in the same solution at room temperature on a shaker for 15 min. Theys were incubated in LC3 primary antibody (MBL, M152-3) diluted 1:500 in PBS/BSA for 1 h at 37 °C. They were then washed 3 × 10 min in PBS/BSA on a shaker, followed by incubation of secondary antibody (Invitrogen Alexa-Fluor 488 goat anti-mouse A21121) diluted in PBS/BSA (1:500) for 30 min at room temperature in the dark. Cells were then washed 3 × 10 min in PBS/BSA before mounting on glass slides using ProLong™ Gold antifade reagent with DAPI (Invitrogen P36935). In Ocy454 Efnb2 knockdown cells, imaging was peformed at ×60 magnification and LC3 puncta were manually counted in five images, containing approximately 15 cells per group; the average per image is reported.

To assess effects of EphrinB2-Fc, EphrinB2-Fc was clustered with immunoglobulin G (IgG) prior to administration for 1 h[10] in the presence or absence of bafilomycin A1, or H1152 (Sigma). We developed an automated method to quantify punctae. Z-stack images were obtained through the full height of the cell layer. Green channel images were Z-projected to gain an overlap of all Z-stack layers, so that all punctae appearing throughout the cell were counted. Channels were split using ImageJ (version 1.48v, National Institutes of Health) and a lower intensity threshold of 40 was used to eliminate non-specific background binding. Cells were manually delineated and punctae with integrated intensity 15–255 were automatically detected and counted in each cell using IMA on MetaMorph (version 7.8.6.0, Molecular Devices); four images were analyzed for each condition in each experiment.

To test the level of autophagy in osteocytes lacking Efnb2, Ocy454 cells with Efnb2 shRNA knockdown and vector controls were differentiated for 11 days and treated with chloroquine (40 μm), a lysosomal degradation inhibitor, for 4 h. Treated and untreated cells were then washed with PBS and lysed in 50 mM Tris, pH 7.4, 150 mM NaCl, 10% glycerol, 1 mM EDTA, 1 mM EGTA, 0.1% NaPyroP, 1% Triton X-100, Roche protease inhibitor. Samples were electrophoresed by 6% sodium dodecyl sulfate-polyacrylamide gel electrophoresis and transferred to Immobilon FL polyvinylidine-flouride membrane (Millipore). Membranes were

blotted with antibodies raised against LC3B (D11, Cell Signaling Technology, 3868), followed by incubation with anti-rabbit IgG secondary antibody fluorescently labelled with IR680 (LI-COR Biosciences). Immunoblots were visualized on an Odyssey membrane imaging system (LI-COR Biosciences) and the LC3-II:I ratio was quantitated as the fold change of EphrinB2quine-treated samples relative to untreated samples.

To assess mineralization, Ocy454 cells with and without Efnb2 knockdown were cultured in the same media as above, plus β-glycerophosphate for 14 days (data shown are with 10 mM, similar results were observed at 2 and 5 mM), stained with Alizarin Red, solubilized, and measured against a standard curve[76].

**Transmission Electron Microscopy**. Femora from female 12-week-old Dmp1Cre and Dmp1Cre.Efnb2$^{f/f}$ mice were fixed for a minimum of 24 h in Karnovsky's fixative. Samples were post-fixed in 1% osmium tetroxide/1.5% potassium ferro-cyanide for 5 h and embedded in Spurr's resin. Ultra-thin sections were stained with uranyl acetate/Reynold's lead citrate and examined with a Philips 300 transmission electron microscope at 60 kV[11]. Osteocyte images were obtained from cells deeply embedded within the bone matrix.

**Statistics**. All graphs show mean ± SEM, SD, or 95% confidence intervals, as indicated in figure legends; number of samples (n) is reported in the figure legends. Statistical significance was determined by unpaired Student's t tests for histomorphometric, mechanical, microCT, and cell culture analyses, and two-way analysis of variance with Fisher's least significant difference test for sFTIRM-derived data (GraphPad Prism 6 (version 6.05)). p < 0.05 was considered statistically significant.

**Reporting summary**. Further information on research design is available in the Nature Research Reporting Summary linked to this article.

## Data availability
RNA-sequencing data generated for this study is deposited with GEO, accession number GSE110795. All other data are available from the corresponding author on request.

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

## Acknowledgements

We thank the staff of the St. Vincent's Health Bioresources Center for excellent animal care and assistance, Mr Joshua Johnson and Mrs Ingrid Poulton for technical assistance with histology, Dr. Roger Curtain (Bio21) for technical assistance with BSEM, Dr. Paul Roschger for advice on BSEM analysis, Dr. Eleftherios Paschalis for advice on collagen crosslinking analysis, and Dr. Elizabeth Allan for work on the Kusa 4b10 cells. This work was supported by NHMRC Grants 1042129 and 1081242 to N.A.S. and T.J.M., Program Grant 1054618 to G.K.S., and a Brockhoff Foundation Grant to C.V. J.S.O. was supported by an ARC Future Fellowship. N.A.S. was supported by an NHMRC Senior Research Fellowship and by the SVI Brenda Shanahan Fellowship. Part of this work was untaken at the Infrared Microspectroscopy Beamline at the Australian Synchrotron, part of ANSTO. C.V. also thanks the Australia and New Zealand Bone and Mineral Society for the award of the Christine & T.J. Martin Travel Award, which allowed the laboratory visit to C.P. St. Vincent's Institute acknowledges the support of the Victorian State Government OIS program.

## Author contributions

Conceptualization, C.V., T.J.M., and N.A.S.; investigation, C.V., T.A.D., N.A., B.C.-I., H.N., Y.H., M.I., G.K.S., N.A.S., L.T., E.J.M., M.B.; data curation, C.V., Y.H., G.K.S., M.B.; analysis, C.V., T.A.D., N.A., C.P., Y.H., L.T., M.B.; writing—original draft, C.V., N.A.S.; writing—review and editing, all authors; visualization, C.V., N.A.S., L.T., M.B.; supervision, N.A.S.; funding acquisition, N.A.S. and T.J.M.

## Additional information

**Competing interests:** The authors declare no competing interests.

