## [Peer Review File · Nature Communications]

Reviewers' Comments:

Reviewer #1:

Remarks to the Author:

This study aims to determine the role of osteocytic ephrinB2 on bone mineralization. In addition to investigating mineralization patterns in the ephrinB2 deficient mice, the authors performed RNA sequencing and showed differential expression of autophagy-associated genes. This led them to conclude that the process of secondary mineralization uses the autophagy machinery in an ephrinB2-dependent manner. The data appear solid and the description is clear. However, the connection between mineralization defects and autophagy is missing, and no attempt was made to determine whether autophagy dysregulation is responsible for the mouse phenotype or whether modulating autophagy at least in vitro has any consequences on the phenotype of ephrinB2-deficient osteocytes. Without this evidence the data on autophagy is not relevant for the manuscript and it should be removed, and the title should be modified accordingly.

Additional issues:

- 1- The first sentence of the introduction should be modified, since the skeleton is unique not only to the human body, and the authors are actually using a mouse model in their studies.
- 2- References should be provided for the statement regarding the process of mineralization in the first paragraph of the Introduction.
- 3- Second paragraph of the introduction, lines 77-79: the statement should be made more general since not only PTHrP, but also PTH (teriparatide) are used to stimulate bone formation.
- 4- Minor point: the reviewer cannot find the "full data availability" statement in the manuscript.

Reviewer #2:

Remarks to the Author:

Following a previous sFTIRM study of bone matrix maturation after PTH treatment, this manuscript aims to define the role of ephrinB2 in osteocytes. It shows that osteocyte-specific ephrinB2-deficient mice have a brittle bone phenotype. It also provides RNA sequencing data using an osteocyte cell line treated with ephrinB2 knockdown vector and reports that a group of autophagy-associated genes were differentially expressed. It proposes that ephrinB2 is required to restrain autophagy in osteocytes and limits the secondary mineralization process.

Although the role of osteocytes in secondary mineralization is an important area of bone biology considering the limited knowledge about the mechanisms controlling the speed of continuing mineral depositions, this study is too preliminary for multiple reasons. First, it is potentially interesting that knockdown of ephrinB2 affects autophagy-related genes, but the mechanism and significance are not clear. The statement was made that autophagic processes in osteocytes directly control mineralization, but there is no such data shown. Second, the high osteocyte lacuna density in the absence of ephrinB2 remains descriptive in nature. Determining the mechanism of this or the causal relationship to a brittle bone phenotype or greater mineral and carbonate deposition would strengthen the manuscript. Third, the conclusions of the FTIR measurements are only poorly supported by the data presented. In particular, time-dependent descriptions such as "brittle bone phenotype is therefore associated with more rapid bone matrix maturation, including rapid accumulation of mineral and carbonate substitution and more rapid collagen compaction within the matrix" (page 9) and "both mineral accrual and carbonate substitution within the mineral occur at an accelerated rate in the absence of ephrinB2" (page 13) appear to be overinterpretation of regional data without performing time-course experiments.

Additional points.

1. Why was a strength defect detectable only in females? Were there any sex differences in other

experiments?

2. Figure 4B and Figure 4E are inconsistent. Wavenumber of Amide I (1588-1712) and Amide II (1500-1600) in Figure 4E is different from the shaded Wavenumber in the FTIRM spectrum (Figure 4B). The photo shown in Figure 4A top was previously published by the authors, but this is not mentioned in the legend. Regions should include endocortical as in Figure 5.
3. The difference between w/w and f/f was observed in periosteal at 0 degree polarization (Figure 5). Why similar differences were not observed at the endosteum or 90 degree polarizing filter?
4. The Load (N)-Deformation (mm) curve and related parameters (Figure 2) are essentially the same information as the Stress (MPa)-Strain (%) curve and related parameters (Figure 3), if the shape of bones were not altered between w/w and f/f as the authors claim.
5. Scale bars are missing from Figure 5A, B, D, and E.
6. Figure 6D. A break in the Y axis is necessary not to start at 0.
7. Does the upregulation of ephrinB2 expression by PTH treatment reduce susceptibility of osteocytes to autophagy?
8. Is there any detectable enhancement of autophagy in osteocyte-specific ephrinB2-deficient mice?

Reviewer #3:

Remarks to the Author:

In this study, the authors found that osteocyte-specific EphrinB2 knockout mice result in fragile bones could cause brittle bone disease. This phenotype was attributed to defect in bone matrix maturation by unregulated secondary mineralization. Further, the authors revealed that autophagy is enhanced in the osteocytes lacking EphrinB2. On the basis of these results, the authors claim that ephrinB2 inhibits autophagy in osteocytes for generating the appropriate mineralization. While the presented data are of interest, some additional experiments are needed to support the author's claim.

Comments

1. The title seems inappropriate since there is no direct evidence that increased autophagy is involved in the pathogenesis of brittle bone.
2. In Fig. 6D, immunofluorescent analysis with anti-LC3 antibody is required (Yoshii SR, Mizushima N. *Int J Mol Sci*. 18. pii: E1865. 2017).
3. The authors should show the results of region 1 and 3 in Fig. 4B.
4. The authors should examine the number and the size of osteocyte projections by staining with fluorescent-tagged phalloidin (Kamioka H. et al. *Bone* 28, 145-149. 2001).
5. The authors should investigate the effects of chloroquine on brittle bone by administration of chloroquine to osteocyte-specific EphrinB2 null mouse.

Responses to Reviewers

Reviewer #1:

This study aims to determine the role of osteocytic ephrinB2 on bone mineralization. In addition to investigating mineralization patterns in the ephrinB2 deficient mice, the authors performed RNA sequencing and showed differential expression of autophagy-associated genes. This led them to conclude that the process of secondary mineralization uses the autophagy machinery in an ephrinB2-dependent manner. The data appear solid and the description is clear. However, the connection between mineralization defects and autophagy is missing, and no attempt was made to determine whether autophagy dysregulation is responsible for the mouse phenotype or whether modulating autophagy at least *in vitro* has any consequences on the phenotype of ephrinB2-deficient osteocytes. Without this evidence the data on autophagy is not relevant for the manuscript and it should be removed, and the title should be modified accordingly.

In response to this comment, we now have new data showing (1) confirmation of increased autophagy in EphrinB2 knockdown osteocytes in vitro (new Figure 7, panels B and C), (2) increased mineralization in cultured EphrinB2-deficient osteocytes (new Figure 7, panels D and E), and (3) abundant autophagosomes and matrix vesicles in EphrinB2 deficient osteocytes in vivo (new Figure 8). This data provides compelling evidence that strongly supports our model proposed in the initial submission. This new data is described at the end of the results section (p12-14), and in some changes to the discussion (p18,19).

Additional issues:

1- The first sentence of the introduction should be modified, since the skeleton is unique not only to the human body, and the authors are actually using a mouse model in their studies.

RESPONSE: We have removed the phrase "in the human body" from this sentence.

2- References should be provided for the statement regarding the process of mineralization in the first paragraph of the Introduction.

RESPONSE: We have added the following two references to this statement:

- Boivin, G. *et al.* Influence of remodeling on the mineralization of bone tissue. *Osteoporosis International* **20**, 1023-1026 (2009), and
- Glimcher, M. J. in *Metabolic Bone Disease and Clinically Related Disorders (Third Edition)* (eds Louis V. Avioli & Stephen M. Krane) 23-52e (Academic Press, 1998).

3- Second paragraph of the introduction, lines 77-79: the statement should be made more general since not only PTHrP, but also PTH (teriparatide) are used to stimulate bone formation.

RESPONSE: We have clarified this statement in the new version of the manuscript, as follows:

"This PTHR1-mediated action is exploited by the pharmacological agents teriparatide (PTH) and abaloparatide (modified N-terminal PTHrP), the only pharmacological agents currently available that can increase bone mass in patients with fragility fractures^{12,13}."

4- Minor point: the reviewer cannot find the “full data availability” statement in the manuscript.

RESPONSE: This has been added to the end of the materials and methods section (p37). The GEO dataset will be made public at the time of publication.

Reviewer #2:

Following a previous sFTIRM study of bone matrix maturation after PTH treatment, this manuscript aims to define the role of ephrinB2 in osteocytes. It shows that osteocyte-specific ephrinB2-deficient mice have a brittle bone phenotype. It also provides RNA sequencing data using an osteocyte cell line treated with ephrinB2 knockdown vector and reports that a group of autophagy-associated genes were differentially expressed. It proposes that ephrinB2 is required to restrain autophagy in osteocytes and limits the secondary mineralization process.

Although the role of osteocytes in secondary mineralization is an important area of bone biology considering the limited knowledge about the mechanisms controlling the speed of continuing mineral depositions, this study is too preliminary for multiple reasons.

First, it is potentially interesting that knockdown of ephrinB2 affects autophagy-related genes, but the mechanism and significance are not clear. The statement was made that autophagic processes in osteocytes directly control mineralization, but there is no such data shown.

RESPONSE: We now have new data showing (1) confirmation of increased autophagy in EphrinB2 knockdown osteocytes (new Figure 7, panels B and C), (2) increased mineralization in cultured EphrinB2-deficient osteocytes (new Figure 7, panels E and F), and (3) abundant autophagosomes and matrix vesicle budding from EphrinB2 deficient osteocytes in vivo (new Figure 8). This compelling new data strongly supports our model proposed in the initial paper, and is described at the end of the results section (p12-14), and in some changes to the discussion (p18,19).

Second, the high osteocyte lacuna density in the absence of ephrinB2 remains descriptive in nature. Determining the mechanism of this or the causal relationship to a brittle bone phenotype or greater mineral and carbonate deposition would strengthen the manuscript.

*RESPONSE: We have not suggested that the high osteocyte lacunar density causes the brittle bone phenotype; such a mechanism would be very difficult to prove. In answer to this reviewer and Reviewer 3, we carried out a Ploton Silver Stain to assess the osteocyte lacunar-canalicular structure; this data is included in **Supplementary Figure 2**. Since this revealed no change in the dendritic network, we have deleted the section of our discussion that suggested autophagy may control osteocyte maturation. We have retained the section describing the increase in osteocyte density, and added a statement that it is not clear whether this is causative or secondary to the greater mineralization (p17).*

Third, the conclusions of the FTIR measurements are only poorly supported by the data presented. In particular, time-dependent descriptions such as “brittle bone phenotype is therefore associated with more rapid bone matrix maturation, including rapid accumulation of mineral and carbonate substitution and more rapid collagen compaction within the matrix” (page 9) and “both mineral accrual and carbonate

substitution within the mineral occur at an accelerated rate in the absence of ephrinB2” (page 13) appear to be overinterpretation of regional data without performing time-course experiments.

RESPONSE: The method we used to measure newly formed periosteal (non-remodelling) bone allows direct comparison of mineralized bone at different stages of maturity. It is, essentially, an in situ time course; this is why we are able to make conclusions about the rate of bone matrix maturation. Bone that is near the surface is less mature than the more deeply embedded bone. We have added further detail in the methods section, and references from us and others, to explain this fully, as follows (p8):

“Unlike the endocortical surface, which is remodeled, the periosteal surface at this location undergoes continuous bone formation (bone growth), without bone resorption²⁰. Measurements taken at increasing depth thereby allow assessment of the rate of bone matrix mineralization as it progresses^{20,21}. Three regions with increasing matrix maturity were measured; data from the control mice confirm the increasing maturity within each region: increasing mineral incorporation (mineral:matrix ratio), increasing carbonate substitution (carbonate:phosphate ratio), and increasing collagen compaction (decreasing amide I:II ratio) (Fig. 4A).”

Additional points.

1. Why was a strength defect detectable only in females? Were there any sex differences in other experiments?

RESPONSE: We have added the following paragraph to the discussion to address this question (p21-22):

*“Sex differences in bone structure, strength, and remodeling exist in all mammalian species including humans and mice^{74,75}, and it is common for genetically altered mice to exhibit sex-specific phenotypes⁷⁶⁻⁷⁸. Both the strength defect and the hypermineralization defect of *Dmp1Cre.Efnb2^{fl/fl}* mice detected here were only observed in female mice. Our earlier work using *Osx1Cre* to initiate deletion of *Ephrin2* in cells earlier in the osteoblast lineage also resulted in a female-specific phenotype¹⁶. However, we found no direct regulation of *Efnb2* in *Ocy454* cells by either estradiol or testosterone in osteocytes (data not shown). We suggest that changes in bone matrix composition may play a more significant role in determining bone strength in the bones of female mice because they have narrower cortices, a higher level of bone remodeling, and lower ultimate strength than male mice⁷⁶⁻⁷⁸.*

2. Figure 4B and Figure 4E are inconsistent. Wavenumber of Amide I (1588-1712) and Amide II (1500-1600) in Figure 4E is different from the shaded Wavenumber in the FTIRM spectrum (Figure 4B). The photo shown in Figure 4A top was previously published by the authors, but this is not mentioned in the legend. Regions should include endocortical as in Figure 5.

RESPONSE: The regions shown in Figure 4B were intended as a guide only, as mentioned in the legend (“approximate regions”). To avoid confusion, we have updated the shading to more accurately represent the peaks used for analysis, and clarified the legend.

We have replaced the image in Figure 4A with a new photo from the present study.

Figure 4 does not include endocortical regions, as these were not analysed by sFTIRM. This is because the endocortical surface, unlike the periosteal surface, undergoes both bone resorption and formation, and is therefore not a region that can be used to study bone

matrix maturation. We have added the following sentence in the results section to explain this (p9): "Unlike the endocortical surface, which is remodeled, the periosteal surface at this location undergoes continuous bone formation (bone growth), without bone resorption²⁰."

3. The difference between w/w and f/f was observed in periosteal at 0 degree polarization (Figure 5). Why similar differences were not observed at the endosteum or 90 degree polarizing filter?

RESPONSE: No difference was observed on the endocortical surface, likely because this region is remodeled, and therefore contains a mix of both mature, and immature bone; we have added a note about this in the results section (p10): "There was no reduction in amide I:II ratio on the endocortical surface; since this region is remodeled, the bone in this region would contain both old and new bone, which may mask any difference".

We have clarified our description of the difference in result between the 0° and 90° polarizing filters on p17 as follows: "Since the reduction in amide I:II was detected under the 0°, but not the 90°, polarizing filter, the increased collagen compaction may be specific to those collagen fibers aligned along the length of the bone, rather than those with radial orientation, or transverse to the bone. This suggests the altered organization of longitudinal fibers may be the main contributor to the ephrinB2-deficient phenotype, and these fibers may be more important for absorbing mechanical forces along the length of the tibiae".

4. The Load (N)-Deformation (mm) curve and related parameters (Figure 2) are essentially the same information as the Stress (MPa)-Strain (%) curve and related parameters (Figure 3), if the shape of bones were not altered between w/w and f/f as the authors claim.

RESPONSE: Yes, and this is why it is important to show both stress-strain and load-deformation data. The load-deformation data (Figure 2) indicates total strength of the bone (without correcting for differences in bone size). It is because we still see a reduction in strength after correcting for bone size (in the stress-strain data – Figure 3), that we know the defect in strength is a material defect. We have added some information to clarify this in the results section (p6-7) and in the figure legends.

5. Scale bars are missing from Figure 5A, B, D, and E.

RESPONSE: Scale bars have been added.

6. Figure 6D. A break in the Y axis is necessary not to start at 0.

*RESPONSE: Thank you for alerting us to this. This has been corrected (now **Fig. 7D**).*

7. Does the upregulation of ephrinB2 expression by PTH treatment reduce susceptibility of osteocytes to autophagy?

RESPONSE: We thank you for this suggestion. We have not tested this and have planned to explore this in the future.

8. Is there any detectable enhancement of autophagy in osteocyte-specific ephrinB2-deficient mice?

*RESPONSE: We have now used transmission electron microscopy to detect autophagy in the osteocyte-specific ephrinB2-deficient mice (**new Figure 8**). This revealed a major morphological change in osteocytes in the *Dmp1Cre.Efnb2^{ff}* mouse model. These cells*

exhibited both a high level of autophagy and a greater level of matrix vesicle formation. This is described in detail in the results section (p13-14), and we have added a number of sentences throughout the discussion to fit this evidence with the model already proposed (p18,19).

Reviewer #3:

In this study, the authors found that osteocyte-specific EphrinB2 knockout mice result in fragile bones could cause brittle bone disease. This phenotype was attributed to defect in bone matrix maturation by unregulated secondary mineralization. Further, the authors revealed that autophagy is enhanced in the osteocytes lacking EphrinB2. On the basis of these results, the authors claim that ephrinB2 inhibits autophagy in osteocytes for generating the appropriate mineralization. While the presented data are of interest, some additional experiments are needed to support the author's claim.

Comments

1. The title seems inappropriate since there is no direct evidence that increased autophagy is involved in the pathogenesis of brittle bone.

RESPONSE: Our title does not claim a direct relationship between increased autophagy and a brittle bone phenotype, but that they are associated.

2. In Fig. 6D, immunofluorescent analysis with anti-LC3 antibody is required (Yoshii SR, Mizushima N. Int J Mol Sci. 18. pii: E1865. 2017).

*RESPONSE: We have now carried this out, and include it in a new **Fig. 7B,C**. This new data confirms that EphrinB2 deficient Ocy454 cells exhibit a high level of endogenous autophagy, to the same level as that observed in control cells stimulated with bafilomycin. This supports our earlier data indicating a high level of autophagy in these cells. This data is described in detail on **p12**.*

3. The authors should show the results of region 1 and 3 in Fig. 4B.

RESPONSE: We have modified this figure to include all three regions.

4. The authors should examine the number and the size of osteocyte projections by staining with fluorescent-tagged phalloidin (Kamioka H. et al. Bone 28, 145-149. 2001).

*RESPONSE: Thank you for this suggestion. We have now assessed the osteocyte network using the Ploton silver stain (the Kamioka method was used in embryonic bone which is less mineralized). We compared 6 different regions, including the diaphysis, the region of mature bone with impaired strength and hypermineralization. We observed no difference in the network. This has been included in the new **Supplementary Figure 2**, and is mentioned in the results section (**p8**). In addition, because this finding indicated no difference in the dendritic network, we have removed the section of our discussion that suggested that osteocyte differentiation and dendrite formation requires EphrinB2.*

5. The authors should investigate the effects of chloroquine on brittle bone by administration of chloroquine to osteocyte-specific EphrinB2 null mouse.

RESPONSE: Systemic inhibition of autophagy with chloroquine would affect all cell types, not only osteocytes. Since systemic treatment with chloroquine has already been noted to reduce osteoclast-mediated bone resorption through direct effects on that lineage (Xiu et

al, JCI 2014), and this would protect bone strength, such an intervention in this mouse model would not be helpful.

Responses to Reviewers

Reviewer #1:

This study aims to determine the role of osteocytic ephrinB2 on bone mineralization. In addition to investigating mineralization patterns in the ephrinB2 deficient mice, the authors performed RNA sequencing and showed differential expression of autophagy-associated genes. This led them to conclude that the process of secondary mineralization uses the autophagy machinery in an ephrinB2-dependent manner. The data appear solid and the description is clear. However, the connection between mineralization defects and autophagy is missing, and no attempt was made to determine whether autophagy dysregulation is responsible for the mouse phenotype or whether modulating autophagy at least *in vitro* has any consequences on the phenotype of ephrinB2-deficient osteocytes. Without this evidence the data on autophagy is not relevant for the manuscript and it should be removed, and the title should be modified accordingly.

In response to this comment, we now have new data showing:

1. *confirmation of increased autophagy in Efnb2 knockdown osteocytes in vitro (new Figure 6, panels B and C),*
2. *increased mineralization in cultured Efnb2 knockdown osteocytes (new Figure 6, panels E and F),*
3. *abundant autophagosomes and matrix vesicles in Dmp1Cre.Efnb2^{f/f} osteocytes in vivo (new Figure 7),*
4. *suppressed autophagy in osteocytes treated with EphrinB2-Fc (new Figure 8, panel E).*

These new data provide compelling evidence that strongly supports our model proposed in the initial submission. It is described at the end of the results section (p12-14), and in some changes to the discussion (p18,19).

Additional issues:

1- The first sentence of the introduction should be modified, since the skeleton is unique not only to the human body, and the authors are actually using a mouse model in their studies.

RESPONSE: We have removed the phrase "in the human body" from this sentence.

2- References should be provided for the statement regarding the process of mineralization in the first paragraph of the Introduction.

RESPONSE: We have added the following two references to this statement:

- Boivin, G. *et al.* Influence of remodeling on the mineralization of bone tissue. *Osteoporosis International* **20**, 1023-1026 (2009), and
- Glimcher, M. J. in *Metabolic Bone Disease and Clinically Related Disorders (Third Edition)* (eds Louis V. Avioli & Stephen M. Krane) 23-52e (Academic Press, 1998).

3- Second paragraph of the introduction, lines 77-79: the statement should be made more general since not only PTHrP, but also PTH (teriparatide) are used to stimulate bone formation.

RESPONSE: We have clarified this statement in the new version of the manuscript, as follows:

“This PTH1R-mediated action is exploited by the pharmacological agents teriparatide (PTH) and abaloparatide (modified N-terminal PTHrP), the only pharmacological agents currently available that can increase bone mass in patients with fragility fractures^{12,13}.”

4- Minor point: the reviewer cannot find the “full data availability” statement in the manuscript.

RESPONSE: This has been added to the end of the materials and methods section (p33). The GEO dataset will be made public at the time of publication.

Reviewer #2:

Following a previous sFTIRM study of bone matrix maturation after PTH treatment, this manuscript aims to define the role of ephrinB2 in osteocytes. It shows that osteocyte-specific ephrinB2-deficient mice have a brittle bone phenotype. It also provides RNA sequencing data using an osteocyte cell line treated with ephrinB2 knockdown vector and reports that a group of autophagy-associated genes were differentially expressed. It proposes that ephrinB2 is required to restrain autophagy in osteocytes and limits the secondary mineralization process.

Although the role of osteocytes in secondary mineralization is an important area of bone biology considering the limited knowledge about the mechanisms controlling the speed of continuing mineral depositions, this study is too preliminary for multiple reasons.

First, it is potentially interesting that knockdown of ephrinB2 affects autophagy-related genes, but the mechanism and significance are not clear. The statement was made that autophagic processes in osteocytes directly control mineralization, but there is no such data shown.

RESPONSE: In our initial manuscript we were very careful to state only that autophagic processes MAY directly control mineralization. Since we did not yet know which types of autophagy are modified, we suggested this as model for how mineralization may be controlled.

We have now carried out additional experiments to provide mechanistic insights to this hypothesis:

- 1. We have confirmed that Efnb2 knockdown increases autophagy using a second method (**new Figure 6, panels B and C**), and now show that EphrinB2-Fc treatment suppresses autophagy (**new Figure 8E**).*
- 2. We have used transmission electron microscopy to identify the autophagic vesicles modified in Dmp1Cre.Efnb2^{fl/fl} osteocytes, and have observed ER degradation, ER-containing autophagosomes and autophagosomes within matrix vesicles in the process of formation on the periphery of these cells (**new Figure 7**).*
- 3. We also now provide mechanistic data showing that the ability of EphrinB2-Fc to stimulate autophagy is RhoA-dependent (**new Figure 8E**). Furthermore, RhoA inhibition, like Efnb2 knockdown, results in increased mineralization (**new Figure 8A-D**).*

These new data strongly support the hypothesis proposed in the initial submission, that EphrinB2 suppresses autophagy, and in its absence autophagic processes are increased and contribute to the process of mineralization. The new data are described at the end of the results section (p11-14), and in some changes to the discussion (p18,19).

Second, the high osteocyte lacuna density in the absence of ephrinB2 remains descriptive in nature. Determining the mechanism of this or the causal relationship to a brittle bone phenotype or greater mineral and carbonate deposition would strengthen the manuscript.

*RESPONSE: We have not suggested that the high osteocyte lacunar density causes the brittle bone phenotype; such a mechanism would be very difficult to prove. In answer to this reviewer and Reviewer 3, we carried out a Ploton Silver Stain to assess the osteocyte lacunar-canalicular structure; this data is included in **Supplementary Figure 2**. Since this revealed no change in the dendritic network, and no change was observed by TEM, we have deleted the section of our discussion that suggested autophagy may control osteocyte maturation. We have retained the section describing the increase in osteocyte density, and added a statement that it is not clear whether this is causative or secondary to the greater mineralization (p17).*

Third, the conclusions of the FTIR measurements are only poorly supported by the data presented. In particular, time-dependent descriptions such as “brittle bone phenotype is therefore associated with more rapid bone matrix maturation, including rapid accumulation of mineral and carbonate substitution and more rapid collagen compaction within the matrix” (page 9) and “both mineral accrual and carbonate substitution within the mineral occur at an accelerated rate in the absence of ephrinB2” (page 13) appear to be overinterpretation of regional data without performing time-course experiments.

RESPONSE: The method we used to measure newly formed periosteal (non-remodelling) bone allows direct comparison of mineralized bone at different stages of maturity. It is, essentially, an in situ time course; this is why we are able to make conclusions about the rate of bone matrix maturation. Bone that is near the surface is less mature than the more deeply embedded bone. We have added further detail in the methods section, and references from us and others, to explain this fully, as follows (p8):

“Unlike the endocortical surface, which is remodeled, the periosteal surface at this location undergoes continuous bone formation (bone growth), without bone resorption²⁰. Measurements taken at increasing depth thereby allow assessment of the rate of bone matrix mineralization as it progresses^{20,21}. Three regions with increasing matrix maturity were measured; data from the control mice confirm the increasing maturity within each region: increasing mineral incorporation (mineral:matrix ratio), increasing carbonate substitution (carbonate:phosphate ratio), and increasing collagen compaction (decreasing amide I:II ratio) (Fig. 3A).”

Additional points.

1. Why was a strength defect detectable only in females? Were there any sex differences in other experiments?

RESPONSE: We have added the following paragraph to the discussion to address this question (p20-21):

“Sex differences in bone structure, strength, and remodeling exist in all mammalian species including humans and mice^{78,79}, and it is common for genetically altered mice to exhibit

sex-specific phenotypes⁸⁰⁻⁸². Both the strength defect and the hypermineralization defect of Dmp1Cre.Efnb2^{f/f} mice detected here were only observed in female mice. Our earlier work using Osx1Cre to initiate deletion of Ephrinb2 in cells earlier in the osteoblast lineage also resulted in a female-specific phenotype¹⁶. However, we found no direct regulation of Efnb2 in Ocy454 cells by either estradiol or testosterone in osteocytes (data not shown). We suggest that changes in bone matrix composition may play a more significant role in determining bone strength in the bones of female mice because they have narrower cortices, a higher level of bone remodeling, and lower ultimate strength than male mice⁸⁰⁻⁸².

2. Figure 4B and Figure 4E are inconsistent. Wavenumber of Amide I (1588-1712) and Amide II (1500-1600) in Figure 4E is different from the shaded Wavenumber in the FTIRM spectrum (Figure 4B). The photo shown in Figure 4A top was previously published by the authors, but this is not mentioned in the legend. Regions should include endocortical as in Figure 5.

RESPONSE: The regions shown in Figure 4B (now Figure 3B) were intended as a guide only, as mentioned in the legend (“approximate regions”). To avoid confusion, we have updated the shading to more accurately represent the peaks used for analysis, and clarified the legend.

We have replaced the image in Figure 4A (now Figure 3A) with a new photo from the present study.

*Figure 4 (now Figure 3) does not include endocortical regions because they were not analysed by sFTIRM. This is because the endocortical surface, unlike the periosteal surface, undergoes both bone resorption and formation, and is therefore not a region that can be used to study bone matrix maturation. We have added the following sentence in the results section to explain this (**p8**): “Unlike the endocortical surface, which is remodeled, the periosteal surface at this location undergoes continuous bone formation (bone growth), without bone resorption²⁰.”.*

3. The difference between w/w and f/f was observed in periosteal at 0 degree polarization (Figure 5). Why similar differences were not observed at the endosteum or 90 degree polarizing filter?

*RESPONSE: No difference was observed on the endocortical surface, likely because this region is remodeled, and therefore contains a mix of both mature, and immature bone; we have added a note about this in the results section (**p10**): “There was no reduction in amide I:II ratio on the endocortical surface; since this region is remodeled, the bone in this region would contain both old and new bone, which may mask any difference”.*

*We have clarified our description of the difference in result between the 0° and 90° polarizing filters on **p16** as follows: “Since the reduction in amide I:II was detected under the 0°, but not the 90°, polarizing filter, the increased collagen compaction may be specific to those collagen fibers aligned along the length of the bone, rather than those with radial orientation, or transverse to the bone. This suggests the altered organization of longitudinal fibers may be the main contributor to the EphrinB2-deficient phenotype, and these fibers may be more important for absorbing mechanical forces along the length of the tibiae”.*

4. The Load (N)-Deformation (mm) curve and related parameters (Figure 2) are essentially the same information as the Stress (MPa)-Strain (%) curve and related parameters (Figure 3), if the shape of bones were not altered between w/w and f/f as the authors claim.

RESPONSE: Yes, and this is why it is important to show both stress-strain and load-deformation data. The load-deformation data (now in Figure 1) indicates total strength of the bone (without correcting for differences in bone size). It is because we still see a reduction in strength after correcting for bone size (in the stress-strain data – now in Figure 2), that we know the defect in strength is a material defect. We have added some information to clarify this in the results section (p6-7) and in the figure legends.

5. Scale bars are missing from Figure 5A, B, D, and E.

RESPONSE: Scale bars have been added (now Figure 4A,B,D,E).

6. Figure 6D. A break in the Y axis is necessary not to start at 0.

*RESPONSE: Thank you for alerting us to this. This has been corrected (now **Fig. 5D**).*

7. Does the upregulation of ephrinB2 expression by PTH treatment reduce susceptibility of osteocytes to autophagy?

RESPONSE: We thank you for this suggestion. We have not tested this and have planned to explore this in the future.

8. Is there any detectable enhancement of autophagy in osteocyte-specific ephrinB2-deficient mice?

*RESPONSE: We have now used transmission electron microscopy to detect autophagy in the osteocyte-specific EphrinB2-deficient mice (**new Figure 7**). This revealed a major morphological change in osteocytes in the *Dmp1Cre.Efnb2^{f/f}* mouse model. These cells exhibit both a high level of autophagy, particularly ER-phagy and a greater level of matrix vesicle formation. This is described in detail in the results section (**p13-14**), and we have added a number of sentences throughout the discussion to fit this evidence with the model already proposed (**p18,19**).*

Reviewer #3:

In this study, the authors found that osteocyte-specific EphrinB2 knockout mice result in fragile bones could cause brittle bone disease. This phenotype was attributed to defect in bone matrix maturation by unregulated secondary mineralization. Further, the authors revealed that autophagy is enhanced in the osteocytes lacking EphrinB2. On the basis of these results, the authors claim that ephrinB2 inhibits autophagy in osteocytes for generating the appropriate mineralization. While the presented data are of interest, some additional experiments are needed to support the author's claim.

Comments

1. The title seems inappropriate since there is no direct evidence that increased autophagy is involved in the pathogenesis of brittle bone.

RESPONSE: Our title does not claim a direct relationship between increased autophagy and a brittle bone phenotype, only that they are both observed in the EphrinB2-deficient osteocytes.

2. In Fig. 6D, immunofluorescent analysis with anti-LC3 antibody is required (Yoshii SR, Mizushima N. Int J Mol Sci. 18. pii: E1865. 2017).

*RESPONSE: We have now carried this out, and include it in a new **Fig. 6B,C**. This new data confirms that EphrinB2-deficient Ocy454 cells exhibit a high level of endogenous autophagy, to the same level as that observed in control cells treated with Bafilomycin. This supports our earlier data indicating a high level of autophagy in these cells. This data is described in detail on **p12**.*

3. The authors should show the results of region 1 and 3 in Fig. 4B.

RESPONSE: We have modified this figure to include all three regions (now Figure 3B).

4. The authors should examine the number and the size of osteocyte projections by staining with fluorescent-tagged phalloidin (Kamioka H. et al. Bone 28, 145-149. 2001).

*RESPONSE: Thank you for this suggestion. We have now assessed the osteocyte network using the Ploton silver stain (the Kamioka method was used in embryonic bone which is less mineralized). We compared 6 different regions, including the diaphysis, the region of mature bone with impaired strength and hypermineralization. We observed no difference in the network. This has been included in the new **Supplementary Figure 2**, and is mentioned in the results section (**p8**) and the discussion (**p13**). Because this finding indicated no difference in the dendritic network, we have also removed the section of our discussion that suggested that osteocyte differentiation and dendrite formation requires EphrinB2.*

5. The authors should investigate the effects of chloroquine on brittle bone by administration of chloroquine to osteocyte-specific EphrinB2 null mouse.

RESPONSE: Systemic inhibition of autophagy with chloroquine would affect all cell types, not only osteocytes. Since systemic treatment with chloroquine has already been noted to reduce osteoclast-mediated bone resorption through direct effects on that lineage (Xiu et al, JCI 2014), and this would protect bone strength by causing high bone mass, such an intervention in this mouse model would not be helpful.

Reviewers' Comments:

Reviewer #1:

Remarks to the Author:

All concerns have been properly addressed

Reviewer #2:

Remarks to the Author:

The authors performed additional experiments to reinforce the autophagy data, which is partly successful. However, the revised manuscript fails because it tries to deliver too much complex information ranging from the altered bone matrix material properties to enhanced autophagy mainly in cultured cell line, which do not really answer to the previous comments of this reviewer.

1 (Previous comment) "The statement was made that autophagic processes in osteocytes MAY directly control mineralization, but there is no such data shown."

The authors claim that new data shows "abundant autophagosomes and matrix vesicles in Dmp1Cre.Efnb2f/f osteocytes in vivo (new Figure 7)" without showing evidence that these structures are matrix vesicles. Are they loaded with calcium and inorganic phosphate ions?

2. (Previous comment) "Second, the high osteocyte lacuna density in the absence of ephrinB2 remains descriptive."

The authors' finding of the greater osteocyte lacunar density in Dmp1Cre.Efnb2f/f bone is interesting and potentially affects mechanical properties of bone. At least, quantitative discussion is necessary whether the higher lacunae density contributes observed differences in bone mechanical properties.

3. (Previous comment) "Third, the conclusions of the FTIR measurements are only poorly supported by the data presented."

The authors' claim that "Unlike the endocortical surface, which is remodeled, the periosteal surface at this location undergoes continuous bone formation (bone growth), without bone resorption (Ref. 20)" is not supported by data.

4. (Previous comment) "Why was a strength defect detectable only in females? Were there any sex differences in other experiments?"

In the revised manuscript, the authors made a very interesting observation that male Dmp1Cre.Efnb2f/f femora showed no significant modification in 222 mineral:matrix, carbonate:mineral or amide I:II ratio (Supplementary Figure 1). This indicates that observed sex difference in mechanical properties is not simply due to geometry differences of bone but due to fundamental differences in bone matrix properties between females and males. Female-male differences in other experiments including autophagy aspects should be further analyzed.

5. (Previous comment) "Does the upregulation of ephrinB2 expression by PTH treatment reduce susceptibility of osteocytes to autophagy?"

This comment is not addressed by the authors.

6. The new scheme Figure 9: Model of osteocytic EphrinB2 regulation of bone matrix composition" is difficult to follow. Why mineral crystals are accumulating far away from autophagic osteocytes.

7. The references are not well selected (too many).

Reviewer #3:

Remarks to the Author:

In the revised manuscript, the authors have addressed and expressed my concerns raised in the previous review. I think the present manuscript is acceptable for publication in Nature Communications.

Both Reviewer 1 and Reviewer 3 requested no further modifications.

Responses to Reviewer 2:

1. COMMENT: The authors claim that new data shows "abundant autophagosomes and matrix vesicles in Dmp1Cre.Efnb2f/f osteocytes in vivo (new Figure 7)" without showing evidence that these structures are matrix vesicles. Are they loaded with calcium and inorganic phosphate ions?

RESPONSE: The presence of calcium and inorganic phosphate ions is not a defining feature of matrix vesicles. Matrix vesicles are simply extracellular membrane-invested particles released by budding from the surfaces of matrix-associated cells, including chondrocytes, osteoblasts, odontoblasts and osteocytes (see HC Anderson, *Clin Orthop Rel Res* 1995). These are clearly observed in Figure 7. We suspect the matrix vesicles in our images contain mineral, but confirming this will be very challenging and requires extensive experiments beyond the scope of this study. We have clarified our statement on p18 to reflect this, as follows (new text in red):

*"We propose that osteocytes also control mineralization by autophagy-dependent release of **matrix vesicles, that may contain** mineral; the observation of many matrix vesicles budding from Dmp1Cre.Efnb2^{f/f} osteocytes, including some containing autophagosomes (Fig. 7), is consistent with this suggestion, **although at this stage, we do not know whether they contain mineral at the time of release.**"*

2. COMMENT: The authors' finding of the greater osteocyte lacunar density in Dmp1Cre.Efnb2f/f bone is interesting and potentially affects mechanical properties of bone. At least, quantitative discussion is necessary whether the higher lacunae density contributes observed differences in bone mechanical properties.

RESPONSE: We are not aware of any data showing that high osteocyte density *per se* modifies bone strength. To discuss this would be highly speculative, and in our opinion, would only lead to confusion.

3. COMMENT: The authors' claim that "Unlike the endocortical surface, which is remodeled, the periosteal surface at this location undergoes continuous bone formation (bone growth), without bone resorption (Ref. 20)" is not supported by data.

RESPONSE: Continuous periosteal bone formation, and endocortical remodelling, are responsible for the expansion of the cortex during growth and in adulthood, and have been well documented (see Seeman, *NEJM*, 2003; Orwoll, *JBMR*, 2003). Our choice of the specific location used for sFTIRM (in Reference 20) was based on many years of studying murine tibial histology in normal mice, where we consistently observe periosteal bone formation at this site. To emphasize this, we have provided Supplementary Figure 2 which shows this in control mice at 6 and 12 weeks of age, and modified the text as follows (new text in red):

*"Unlike the endocortical surface, which is remodeled, the periosteal surface is a site of **modeling-based cortical expansion** and undergoes continuous bone formation (bone growth)¹⁶. **We have confirmed this in control mice at the site used for sFTIRM analysis at both 6 and 12 weeks of age (Supplementary Figure 2)**".*

4. COMMENT: In the revised manuscript, the authors made a very interesting observation that male Dmp1Cre.Efnb2f/f femora showed no significant modification in

mineral:matrix, carbonate:mineral or amide I:II ratio (Supplementary Figure 1). This indicates that observed sex difference in mechanical properties is not simply due to geometry differences of bone but due to fundamental differences in bone matrix properties between females and males. Female-male differences in other experiments including autophagy aspects should be further analyzed.

RESPONSE: We agree that the difference in mechanical properties is unlikely to be due to geometry differences, but is due to baseline differences in bone matrix composition between females and males. Given that the male *Dmp1Cre.Efnb2f/f* femora show no strength phenotype, it would be very difficult to justify the use of additional animals to specifically validate whether they also exhibit no change in autophagy in their osteocytes. We have clarified our comment about sex differences in the discussion, as follows: *"We suggest that changes in bone matrix composition may play a more significant role in determining bone strength in female bones because they have higher mineral content and more rapid bone remodeling than male bones."*

5. COMMENT: "Does the upregulation of ephrinB2 expression by PTH treatment reduce susceptibility of osteocytes to autophagy?". This comment is not addressed by the authors.

RESPONSE: As we stated in the previous response, we plan to assess the effects of PTH treatment on autophagy. This requires an extensive set of *in vivo* experiments, which is beyond the scope of the current work.

Given that pharmacologic PTH treatment does not modify the mineralization process (Vrahnas et al, Bone 2016), we had commented in the discussion that, in a physiological context, any PTH1R-mediated action on autophagy is more likely to relate to actions of endogenous PTHrP. See p20, end of first paragraph, included here for convenience:

"Since exogenous PTH and endogenous PTHrP promote EphrinB2 expression in both osteoblasts and osteocytes, they may promote both initiation of osteoid mineralization and restrain the rate of mineral accrual. While there is no evidence for altered mineral accrual with PTH pharmacological administration, the reduced material strength of mice with osteocyte-specific PTHrP deletion suggests PTHrP may regulate physiological mineral accrual by inhibiting autophagy through EphrinB2-dependent actions in the osteocyte."

6. COMMENT: The new scheme Figure 9: Model of osteocytic EphrinB2 regulation of bone matrix composition" is difficult to follow. Why mineral crystals are accumulating far away from autophagic osteocytes.

RESPONSE: We have simplified the diagram, modified the legend for clarity, and changed the labelling to make it clear that the crystals are not accumulating far from the osteocytes, but likely within their lacunar space.

7. COMMENT: The references are not well selected (too many).

RESPONSE: We have removed references where possible (17 references removed).

Reviewers' Comments:

Reviewer #2:

Remarks to the Author:

I felt that the possible link between autophagy and mineralization is not yet robust. For example, I was not convinced with "abundant extrusions and matrix vesicles budding from the cell surface" (line 317). Are these vesicles really associated with autophagy and involved in mineralization? This is important because osteoblasts/osteocytes under non-mineralizing conditions also produce exosome-like vesicles containing bio-molecules including miRNAs (ref. 1).

Mineralization in matrix vesicles can be evaluated by analyzing composition of Ca, P, and C with scanning electron microscopy (SEM) with energy-dispersive X-ray spectroscopy (EDS, EDX or EDXS), which is a widely used analytical technique (ref. 2). Transmission electron microscopy (TEM)-EDS (ref. 3) may be another option using preparations for TEM (Figure 7). Various other sophisticated methods to analyze minerals in matrix vesicles have also been reported (ref. 4).

I would encourage the authors to reinforce the autophagy-mineralization link further. Alternatively, the authors should tone down the statement about matrix vesicles and by using other more general terms such as extracellular vesicles etc.

References

1. Choi SY, et al. Regulating Osteogenic Differentiation by Suppression of Exosomal microRNAs. *Tissue Eng Part A*. 2018. doi: 10.1089/ten.TEA.2018.0257.
2. de Faria AN, et al. Estrogen and phenol red free medium for osteoblast culture: study of the mineralization ability. *Cytotechnology*. 2016;68(4):1623-32. doi: 10.1007/s10616-015-9844.
3. Bozycki L, et al. Analysis of Minerals Produced by hFOB 1.19 and Saos-2 Cells Using Transmission Electron Microscopy with Energy Dispersive X-ray Microanalysis. *J Vis Exp*. 2018;(136). doi: 10.3791/57423.
4. Boonrungsiman S, et al. The role of intracellular calcium phosphate in osteoblast-mediated bone apatite formation. *Proc Natl Acad Sci U S A*. 2012;109(35):14170-5. doi: 10.1073/pnas.1208916109.

Response to Reviewer #2:

COMMENT 1: I felt that the possible link between autophagy and mineralization is not yet robust. For example, I was not convinced with “abundant extrusions and matrix vesicles budding from the cell surface” (line 317). Are these vesicles really associated with autophagy and involved in mineralization? This is important because osteoblasts/osteocytes under non-mineralizing conditions also produce exosome-like vesicles containing bio-molecules including miRNAs (ref. 1).

*RESPONSE: To clarify, we did not intend to suggest that the extrusions and matrix vesicles are evidence of autophagy – they are evidence of release of matrix vesicles. **We have modified the text to clarify this (p12-13).***

Our in vivo evidence for increased autophagy in the hypermineralized bone is the presence of high numbers of autophagosomes in the EphrinB2-deficient osteocytes. The high level of autophagy in response to EphrinB2 deficiency was also detected in vitro in Ocy454 osteocytes with EphrinB2 knockdown, which exhibit a high level of mineralization. It was also confirmed when EphrinB2-Fc treatment of osteocytes suppressed autophagy.

COMMENT 2: Mineralization in matrix vesicles can be evaluated by analyzing composition of Ca, P, and C with scanning electron microscopy (SEM) with energy-dispersive X-ray spectroscopy (EDS, EDX or EDXS), which is a widely used analytical technique (ref. 2). Transmission electron microscopy (TEM)-EDS (ref. 3) may be another option using preparations for TEM (Figure 7). Various other sophisticated methods to analyze minerals in matrix vesicles have also been reported (ref. 4).

RESPONSE: The new experiments proposed by the referee seek to answer additional questions about the nature of matrix vesicle mineral crystals from EphrinB2-deficient osteocytes. We are concerned that the methods used in the quoted studies are all limited to cultured osteoblasts, where studying mechanisms of mineral release is still highly controversial because cultured cells do not faithfully replicate the 3 dimensional structure of lamellar bone, and high levels of phosphate must be provided to induce mineral deposition. This is particularly troublesome in cultured osteocytes, where methods are still in their infancy. Cultured osteocytes do not produce collagen-containing mineral nodules, but produce a diffuse mineral layer (see Figure 6F), and methods for culturing osteocytes within a mineralized collagen matrix have not yet been developed. This distinction is important because primary mineralization is not modified in the EphrinB2-deficient mouse. Studying the cultured osteocytes by current methods will not provide information on the changes in mineral content of matrix vesicles during secondary mineralization, where they interact with a mineralized collagen matrix. We cannot use osteoblasts for these questions because they are not embedded in the matrix, and because EphrinB2 deletion in osteoblasts leads to apoptosis (Tonna et al, 2014).

It should also be noted that reference 2 (cited by the referee) measures Ca/P ratio of entire nodules (not vesicles). Although the referee implied that the method for reference 3 in cultured cells could be applied to our TEM preparations, this is not correct. Due to its high level of mineralization, TEM analysis of adult bone (as opposed to cultured cells) requires decalcification, which unfortunately, removes the mineral we would wish to analyse.

We have added a section to the discussion describing limitations of current osteocyte culture methods and the requirement for developing systems where osteocytes are embedded in a mineralized collagen-containing matrix (p19).

COMMENT 3: I would encourage the authors to reinforce the autophagy-mineralization link further. Alternatively, the authors should tone down the statement about matrix vesicles and by using other more general terms such as extracellular vesicles etc.

RESPONSE: In order to reinforce the autophagy-mineralization link we needed first to identify the autophagic pathway that is involved so that we can specifically inhibit it. Our initial RNAseq data suggested that the type of autophagy modified in the bones of these mice was not canonical degradative macro-autophagy (which involves changes in Atg proteins and can be genetically and pharmacologically manipulated), but was likely to be mitophagy or ER-phagy. Our TEM imaging, added in the first revision, indeed showed that autophagosomes in EphrinB2-deficient osteocytes contain degraded endoplasmic reticulum, indicating a specific increase in ER-phagy. To test whether the mineral release is ER-phagy-dependent will require selective modification of this process. However, at this stage there are no pharmacological reagents available that can specifically stimulate ER-phagy in vitro or in vivo, nor has anyone yet developed a genetic approach to stimulate ER-phagy in vivo in a cell-specific manner. This is why we have focussed our attention on identifying the signalling pathway (RhoA) that is involved.

We have added a section to the discussion to clarify that there is no association with increased Atg-dependent autophagy, but with ER-phagy, and to inform readers that the direct role of ER-phagy in mineral release is not something that can yet be tested experimentally (p18, first paragraph).

Regarding the request to change our description to refer to extracellular vesicles rather than matrix vesicles, we return to the original definition of matrix vesicles. Matrix vesicles were discovered by H Clarke Anderson and defined as follows (Anderson HC, Current Rheumatology Reports, 2003): "Matrix vesicles (MVs) are extracellular, 100 nm in diameter, membrane-invested particles selectively located at sites of initial calcification in cartilage, bone, and pre-dentin.". Simply put, matrix vesicles are extracellular vesicles released by cells residing at sites of mineralization, and we have shown the presence of these vesicles in the bone matrix in the TEM images (added in the 1st revision). Matrix vesicles do not necessarily contain mineral at the time of release (Anderson HC, Clinical Orthopedics and Related Research, 1995). This is the controversial question being addressed by the Boonrungsiman paper (reference 3 by referee), which was limited to in vitro studies. For this reason, we do not see a need to rectify our statement that these are matrix vesicles.

We have modified the discussion to include the definition of matrix vesicles, and describe the current controversies regarding the mechanisms of mineral release (p18 second paragraph). We have also modified the final schematic figure (panel C) to omit the mineral within the vesicles prior to release, since is a hypothesis arising from the work.

Reviewers' Comments:

Reviewer #2:

Remarks to the Author:

The manuscript has improved from previous submissions. Although many questions are left unanswered, this work makes a significant contribution to the field.

Response to Reviewer #2:

COMMENT 1: I felt that the possible link between autophagy and mineralization is not yet robust. For example, I was not convinced with “abundant extrusions and matrix vesicles budding from the cell surface” (line 317). Are these vesicles really associated with autophagy and involved in mineralization? This is important because osteoblasts/osteocytes under non-mineralizing conditions also produce exosome-like vesicles containing bio-molecules including miRNAs (ref. 1).

*RESPONSE: To clarify, we did not intend to suggest that the extrusions and matrix vesicles are evidence of autophagy – they are evidence of release of matrix vesicles. **We have modified the text to clarify this (p12-13).** Our in vivo evidence for increased autophagy in the hypermineralized bone is the presence of high numbers of autophagosomes in the EphrinB2-deficient osteocytes.*

COMMENT 2: Mineralization in matrix vesicles can be evaluated by analyzing composition of Ca, P, and C with scanning electron microscopy (SEM) with energy-dispersive X-ray spectroscopy (EDS, EDX or EDXS), which is a widely used analytical technique (ref. 2). Transmission electron microscopy (TEM)-EDS (ref. 3) may be another option using preparations for TEM (Figure 7). Various other sophisticated methods to analyze minerals in matrix vesicles have also been reported (ref. 4).

I would encourage the authors to reinforce the autophagy-mineralization link further. Alternatively, the authors should tone down the statement about matrix vesicles and by using other more general terms such as extracellular vesicles etc.

RESPONSE: The new experiments proposed by the referee seek to answer additional questions about the nature of matrix vesicle mineral crystals from EphrinB2-deficient osteocytes. Please note that reference 2 (cited by the referee) measures Ca/P ratio of entire nodules (not vesicles). The methods used in the quoted studies are all limited to cultured osteoblasts, and as such are likely to reflect primary mineralization, which is not modified in the EphrinB2-deficient mouse. Cultured osteocytes do not produce collagen-containing mineral nodules, but produce a diffuse mineral layer (see Figure 6F), and methods for culturing osteocytes within a mineralized collagen matrix are still being developed. Although the referee implied that the method for reference 3 in cultured cells could be applied to our TEM preparations, this is not correct. Due to its high level of mineralization, TEM analysis of adult bone (as opposed to cultured cells) requires decalcification, which unfortunately, removes the mineral we would wish to analyse.

*Given the limitations of current cell culture systems available, to reinforce a direct link between autophagy in osteocytes and mineralization we have treated Ocy454 osteocytes with the autophagy stimulus rapamycin for 24 hours. To mimic secondary mineralization as closely as possible, we differentiated the cells until mineralization had commenced before initiating treatment. In the rapamycin-treated cells, a higher level of mineralization, approximately double that of control, was detected by Alizarin Red staining. This provides firm proof-of-principle evidence that promoting autophagy directly stimulates secondary mineralization by osteocytes. **We have added this new data to Figure 6G and included details about it in the methods, results, and discussion.***

*Regarding the request to change our description to refer to extracellular vesicles rather than matrix vesicles, we return to the original definition of matrix vesicles. Matrix vesicles were discovered by H Clarke Anderson and defined as follows (Anderson HC, Current Rheumatology Reports, 2003): "Matrix vesicles (MVs) are extracellular, 100 nm in diameter, membrane-invested particles selectively located at sites of initial calcification in cartilage, bone, and predentin.". Simply put, matrix vesicles are extracellular vesicles released by cells residing at sites of mineralization, and we have shown the presence of these vesicles in the bone matrix in the TEM images (added in the 1st revision). Matrix vesicles do not necessarily contain mineral at the time of release (Anderson HC, Clinical Orthopedics and Related Research, 1995). This is the controversial question being addressed by the Boonrungsiman paper (reference 3 by referee), which was limited to in vitro studies in osteoblasts. For this reason, we do not see a need to rectify our statement about matrix vesicles. **We have modified the discussion to include the definition of matrix vesicles, and describe the current controversies regarding the mechanisms of mineral release (p18 second paragraph). We have also modified the final schematic figure (panel C) to omit the mineral within the vesicles prior to release, since is a hypothesis arising from the work.***

Reviewers' Comments:

Reviewer #2:

Remarks to the Author:

The additional in vitro experiment shown in Fig. 6G is to demonstrate that pharmacologically induced autophagy in mineralizing Ocy454 osteocytes dramatically (more than twofold) increases Alizarin Red area within 24 hrs.

1. In all other in vitro mineralization experiments, Alizarin Red (uM) was quantified (Fig. 6E, Fig. 8B, D). Alizarin Red (uM) should be shown rather than Alizarin Red area (% total), because change in cell shape such as spreading can affect the ImageJ area measurement. Numerous papers suggest that actin cytoskeleton and cell-cell adhesion are the major targets of rapamycin. Please show Alizarin Red (uM) instead of Alizarin Red area (% total).

2. Yang et al (2015) reported effects of rapamycin treatment on osteoblasts in a rat model of bone fracture. Is this consistent with the current proof-of-principle experiment? If rapamycin "leads to a bone fragility" through autophagy of osteocytes, could this be a potential limitation of rapamycin therapy in the treatment of certain human diseases.

Yang GE, Duan X, Lin D, Li T, Luo D, Wang L, et al. Rapamycin-induced autophagy activity promotes bone fracture healing in rats. *Exp Ther Med.* 2015; 10:1327–33.

Response to reviewer:

REVIEWER'S QUERY: The additional in vitro experiment shown in Fig. 6G is to demonstrate that pharmacologically induced autophagy in mineralizing Ocy454 osteocytes dramatically (more than twofold) increases Alizarin Red area within 24 hrs. 1. In all other in vitro mineralization experiments, Alizarin Red (uM) was quantified (Fig. 6E, Fig. 8B, D). Alizarin Red (uM) should be shown rather than Alizarin Red area (% total), because change in cell shape such as spreading can affect the ImageJ area measurement. Numerous papers suggest that actin cytoskeleton and cell-cell adhesion are the major targets of rapamycin. Please show Alizarin Red (uM) instead of Alizarin Red area (% total).

OUR RESPONSE: Measuring Alizarin Red area for mineralization is a standard method, commonly used in the field (see Perpetuo, Bourne & Orris, *Bone Research Protocols*, 2019, reference 69, added to the manuscript). While we solubilized Alizarin Red for our earlier experiments, we changed our procedure for the new experiment because these cells had a greater tendency than usual to “curl up” at the edge of the confluent layer (see image below, provided for the reviewer). As you can see, Alizarin Red stain gets trapped in those folded areas, and when solubilized would overestimate the amount of mineralization and skew results depending on how much folding has occurred. To overcome this, we measured mineralised area in the largest circular region possible at the centre of the well (shown by a dashed line in the image). This is a standard method.

To clarify the reason for this change, we have added the following detail to the methods section (highlighted in red in the submitted manuscript): *Since folding of the cell layer occurred at the edge of the wells and trapped Alizarin Red stain, mineralization was quantified using areal thresholding in a circular region (avoiding the folded areas) with ImageJ; values are expressed as the percentage of the area measured, as previously described⁷¹.*

With respect to the claim that cell spreading itself may alter mineralisation area, this is highly unlikely, since Ocy454 cells are confluent, mineral is deposited external to the cells, and both area (and solubilized Alizarin Red) are measured across the entire well.

REVIEWER'S QUERY: 2. Yang et al (2015) reported effects of rapamycin treatment on osteoblasts in a rat model of bone fracture. Is this consistent with the current proof-of-principle experiment? If rapamycin “leads to a bone fragility” through autophagy of osteocytes, could this be a potential limitation of rapamycin therapy in the treatment of certain human diseases.

Yang GE, Duan X, Lin D, Li T, Luo D, Wang L, et al. Rapamycin-induced autophagy activity promotes bone fracture healing in rats. *Exp Ther Med*. 2015;10:1327–33.

OUR RESPONSE: We used rapamycin as an *in vitro* stimulus of autophagy and would not recommend its use as a therapeutic approach. Treating patients systemically with rapamycin would influence every cell in the body. Additionally, we recently reported that rapamycin treatment compromises bone growth, thereby reducing bone strength (Bateman, et al. reference 50 in the revised manuscript). As the reviewer suggests, there may also be a direct effect on osteocytes that limits bone strength. We have added a few sentences to clarify this point in the discussion (p18, highlighted in red in the submitted manuscript), as follows:

Whether rapamycin treatment in vivo would increase bone stiffness by stimulating mineralization is not known, and effects on other cell types are likely to mask such action. We recently tested rapamycin treatment in wild type mice and in the $\alpha 2(1)$ -G610C model of osteogenesis imperfecta⁵⁰. Although rapamycin increased trabecular bone mass (consistent with a previous report where it increased fracture callus size⁵¹) such treatment also reduced cortical bone growth, thereby reducing bone strength⁵⁰. An increase in mineralization of bone deposited during the rapamycin treatment period may have also played a part in weakening the bone structure.

Reviewers' Comments:

Reviewer #2:

Remarks to the Author:

I understood that "the largest circular region possible at the center of the well" was used for the measurement. This is fine. This region should be used to measure Alizarin Red [μM]. The value normalized to the area would be strengthen the paper.

Response to Reviewer 2:

Reviewer #2 (Remarks to the Author):

I understood that “the largest circular region possible at the center of the well” was used for the measurement. This is fine. This region should be used to measure Alizarin Red [uM]. The value normalized to the area would be strengthen the paper.

RESPONSE: Removing and solubilizing the central region of cells in the Alizarin Red assay is not technically feasible and is not an accepted method for assessing mineralization. To attempt to reproducibly remove a consistent central region for solubilization would need to be done physically, and is not possible. This is because Ocy454 cells, like all osteoblast lineage cells, are confluent adherent cells. After fixation and Alizarin Red staining, they are tightly attached to the tissue culture plastic, and could only be removed by scraping the dried adherent cells into another vessel. To attempt to remove a reproducible central region would introduce unacceptable errors in processing. Both methods we have used are well accepted and well validated methods for quantifying mineralization by cultured cells.

It's worth noting that it is the Alizarin Red stain that is solubilized, and not the cells themselves. The Alizarin Red is bound to extracellular calcium deposited by the cells. With the folding that occurred in these wells, the stain was trapped in the folds. While this interferes with the imaging, it also interferes when it is solubilised because there is no way to distinguish between the stain that was on the cells and the stain that was in the folds; if you took the whole well, you would have both the stain within the fold and the stain on the cells. In these wells, there is almost as much stain within the folds, so it would introduce an unacceptable level of noise (possibly more noise than signal). This is why many laboratories use image analysis rather than solubilisation.